# Analysis of context-specific KRAS–effector (sub) complexes in Caco-2 cells

Camille Ternet[1,2,*], Philipp Junk[1,2,*], Thomas Sevrin[1,2,*], Simona Catozzi[1,2], Erik Wåhlén[4], Johan Heldin[4], Giorgio Oliviero[1], Kieran Wynne[1,3], Christina Kiel[5,1,2]

**Ras is a key switch controlling cell behavior. In the GTP-bound form, Ras interacts with numerous effectors in a mutually exclusive manner, where individual Ras–effectors are likely part of larger cellular (sub)complexes. The molecular details of these (sub)complexes and their alteration in specific contexts are not understood. Focusing on KRAS, we performed affinity purification (AP)–mass spectrometry (MS) experiments of exogenously expressed FLAG-KRAS WT and three oncogenic mutants ("genetic contexts") in the human Caco-2 cell line, each exposed to 11 different culture media ("culture contexts") that mimic conditions relevant in the colon and colorectal cancer. We identified four effectors present in complex with KRAS in all genetic and growth contexts ("context-general effectors"). Seven effectors are found in KRAS complexes in only some contexts ("context-specific effectors"). Analyzing all interactors in complex with KRAS per condition, we find that the culture contexts had a larger impact on interaction rewiring than genetic contexts. We investigated how changes in the interactome impact functional outcomes and created a Shiny app for interactive visualization. We validated some of the functional differences in metabolism and proliferation. Finally, we used networks to evaluate how KRAS–effectors are involved in the modulation of functions by random walk analyses of effector-mediated (sub)complexes. Altogether, our work shows the impact of environmental contexts on network rewiring, which provides insights into tissue-specific signaling mechanisms. This may also explain why KRAS oncogenic mutants may be causing cancer only in specific tissues despite KRAS being expressed in most cells and tissues.**

## Introduction

The interactome of a cell, like a social network, refers to the entirety of interactions of cellular molecules, in particular, protein–protein interactions (PPIs) (Vidal et al, 2011). These interactions form a network and impact the spatial protein localization and functional organization of a cell. Networks adapt to internal and external cues by converting the signals in responses to stimuli into a plethora of possible output functions that drive cell fates and phenotypes. PPIs, as the core of signaling networks, impact how signals are transduced, and alterations in cellular networks are often linked to diseases, particularly complex diseases such as cancer (van Boxel-Dezaire et al, 2006; Vidal et al, 2011). Mutations in oncogenes can perturb PPI networks (Hammond et al, 2015) when protein catalytic and binding functions are affected resulting in alterations in the proteins' binding interfaces (Kiel & Serrano, 2014).

The oncoprotein KRAS is an example of a hub signaling protein, as it is part of a highly interconnected and dynamic network capable of interacting with many other proteins (Kiel et al, 2021). Oncogenic mutations in KRAS rewire interactions and signaling pathways (Kennedy et al, 2020). KRAS belongs to the Ras superfamily of GTPases and acts as a molecular switch that cycles between an inactive GDP-bound state and an active GTP-bound state. The GTP-bound Ras protein mediates binding to several downstream proteins, thereby controlling essential and diverse cellular processes such as survival, polarization, proliferation, differentiation, apoptosis, and migration (Simanshu et al, 2017; Ibáñez Gaspar et al, 2021). It is still enigmatic how Ras does all of it. However, it is known that a class of proteins called "effectors" plays a critical role (Kiel et al, 2013, 2021; Gimple & Wang, 2019).

Ras–effectors are defined as proteins that bind much stronger (i.e., with higher affinity or lower $K_d$ value) to Ras·GTP than to Ras·GDP. Their interaction with Ras·GTP relies on a domain with a ubiquitin-like topology of three types: the Ras-binding domain (RBD), the Ras association (RA) domain, or the PI3K_rbd, which will herein collectively be referred to as RBDs. All effector RBDs recognize the same switch regions of Ras·GTP, which results in mutually exclusive binding (Shields et al, 2000; Ibáñez Gaspar et al, 2021). Although the presence of an RBD is a necessary condition to qualify as an effector for Ras·GTP, it is not a sufficient criterion.

[1]Systems Biology Ireland, School of Medicine, University College Dublin, Dublin 4, Ireland   [2]UCD Charles Institute of Dermatology, School of Medicine, University College Dublin, Dublin 4, Ireland   [3]Conway Institute of Biomolecular & Biomedical Research, University College Dublin, Dublin 4, Ireland   [4]Department of Pharmaceutical Biosciences, Uppsala University, Uppsala, Sweden   [5]Department of Molecular Medicine, University of Pavia, Pavia, Italy

Correspondence: christina.kiel@unipv.it
*Camille Ternet, Philipp Junk, and Thomas Sevrin contributed equally to this work

Indeed, for a total of 56 effectors that contain RBDs, the binding affinities between Ras·GTP and effector complexes are known (either from experiments or from computational predictions) to vary, and some are predicted not to bind at all (Fig S1) (Kiel et al, 2005; Wohlgemuth et al, 2005; Ibáñez Gaspar et al, 2021; Kiel et al, 2021; Rezaei Adariani et al, 2021).

In addition to affinities between the RBDs and Ras·GTP, protein abundance is important for complex formation. In a previous study, we used protein abundance together with binding affinities in a mathematical model to predict the amount of each of the 56 effectors in complex with Ras·GTP in 29 human tissues (Catozzi et al, 2021). Surprisingly, only nine effectors form significant complexes (≥5%) with Ras·GTP in at least one of the 29 tissues (here referred to as group 1 effectors). This raised the question about the relevance of the remaining effectors, some of which are well-established effectors such as PI3-kinase (PI3K) (Castellano & Downward, 2011). As effectors are generally multidomain proteins, we reasoned that domains that can transfer effectors to the plasma membrane (PM), where Ras·GTP is localized, can increase the number of complexes formed between Ras·GTP and effectors (Ibáñez Gaspar et al, 2021). Indeed, seminal work by Kholodenko and colleagues has demonstrated that membrane anchoring of both interacting proteins strongly increases the average lifetime of complexes, that is, the "piggyback" mechanism (Kholodenko et al, 2000). When we applied the piggyback mechanism to the Ras–effector model, we identified 32 effectors that are predicted to form significant complexes with Ras·GTP only with an additional domain recruited to the PM (here referred to as group 2 effectors) (Fig S1). These effectors were predicted to be recruited to the PM in response to specific conditions (e.g., inputs/stimuli/growth factors) (Ibáñez Gaspar et al, 2021; Kiel et al, 2021). The remaining 15 effectors are never predicted to be in significant complex with Ras·GTP and are likely no true Ras–effectors (here referred to as group 3 effectors).

Colorectal cancer (CRC) is the fourth leading cause of cancer death worldwide. In 2018, there were 1.8 million new CRC cases reported, with a significant shift from older to younger individuals (Siegel et al, 2020). CRC develops through a complex sequence of processes involving an accumulation of epigenetic and genetic alterations, where the major drivers appear to be KRAS mutations and specific pathways that regulate cell growth and differentiation (Fearon & Vogelstein, 1990). The most frequent KRAS mutations found in CRC are single-point mutations found at codon 12 (i.e., G12D, G12V, and G12C) followed by codons G13 and Q61 (Hobbs et al, 2016; Tate et al, 2019). Oncogenic KRAS leads to an accumulation of constitutively active (GTP-bound) KRAS proteins leading toward the activation of diversified downstream signaling pathways such as the Ras/RAF/MEK/ERK signaling pathway and the PI3K/AKT signaling pathways, which were extensively studied in the Ras–effector cancer context (Romano et al, 2014). However, there is evidence that other Ras–effectors play a role in cancer (Engin et al, 2017).

In this work, we experimentally probed context-specific network rewiring of KRAS exogenously expressed with a FLAG-tag in immortalized human Caco-2 cells. This cell line, derived from human colorectal adenocarcinoma cells, harbors somatic APC mutations and CTNNB1 (i.e., β-catenin) mutations, but is WT for KRAS (Fogh et al, 1977). To probe different "genetic contexts," we exogenously expressed KRAS WT and three oncogenic mutations frequently found in CRC (G12V, G12D, and G12C) to probe different "genetic contexts." To probe different "culture contexts," we grew Caco-2 cells in various growth media mimicking tumor microenvironments (TME) that are known to impact CRC maintenance, progression, and metastasis, which have been described earlier in connection with oncogenic KRAS. IL-6 and TNF-α, both being part of the inflammatory response found in the TME, are factors of those growth culture contexts (Ancrile et al, 2007; Waldner et al, 2012; Zeng et al, 2017). (Patho)physiological conditions such as hypoxia (Kikuchi et al, 2009; Chun et al, 2010) (mimicked by dimethyloxalylglycine, DMOG [Zhang et al, 2016]), EGF, and PGE2 also play a role in CRC and KRAS TME (Smakman et al, 2005; Greenhough et al, 2009; Hsu et al, 2017) and were selected as growth conditions here. Each combination of genetic and culture contexts was analyzed separately in affinity purification–mass spectrometry (AP-MS) experiments (Hein et al, 2015; Richards et al, 2021) to determine KRAS-mediated complexes. Our study provides an in-depth reconstruction of PPI networks mediated by oncogenic KRAS–effector proteins in culture contexts that mimic some aspects of (patho)physiological colon contexts. In addition, by identifying different levels of network organization called subcomplexes, we further detailed the downstream pathways mediated by effectors of KRAS and linked them to functional outputs.

# Results

## Analysis of KRAS-mediated networks in different genetic and culture contexts

We conducted AP-MS experiments to characterize the PPI landscape of both the WT and oncogenic mutant forms of KRAS in different growth conditions. KRAS protein variants were exogenously expressed as FLAG-tagged proteins under the control of a doxycycline-inducible promoter (Beltran-Sastre et al, 2015). As observed earlier (Beltran-Sastre et al, 2015), the promoter shows some leakiness even without doxycycline. As we aimed to express FLAG-KRAS at relatively physiological levels, doxycycline was only added to express the FLAG-KRAS WT proteins at a dose that resulted in equal expression levels compared with the FLAG-KRAS mutant proteins expressed without doxycycline (Fig S2).

To analyze the KRAS WT and mutant interactomes in different growth media ("culture contexts") that mimic conditions relevant in the colon and CRC, Caco-2 cells were grown 4 h after transfection for 24 h in minimal medium (DMEM with 2 mM l-glutamine) supplemented with either IL-6, TNF-α, PGE2, EGF, or the HIF-hydroxylase inhibitor DMOG at different concentrations (20 and 200 ng/ml) before the AP-MS experiment was conducted. The expression levels of both KRAS (Fig S3) and effectors (Fig S4) were generally in a similar order of magnitude when grown in different "culture contexts" (assayed by Western blotting and MS analysis of whole-cell lysates before the AP-MS experiment). Altogether, we tested four "genetic contexts" (FLAG-KRAS WT, G12V, G12D, and G12C) and 11 "culture contexts" (minimal medium, and two concentrations each of IL-6, TNF-α, PGE2, EGF, and DMOG in minimal medium), resulting in 44 condition-specific AP-MS experiments (Fig 1).

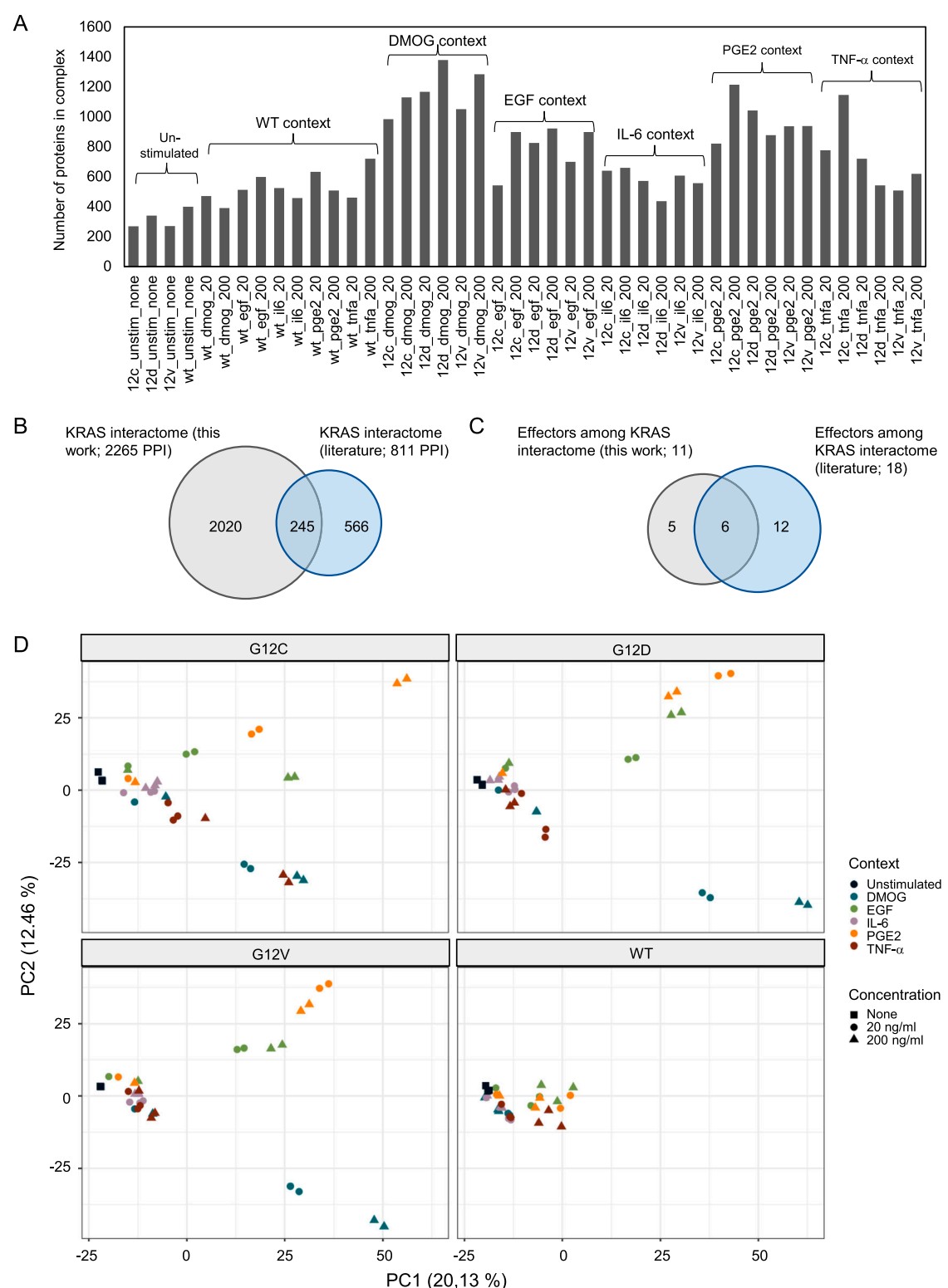

**Figure 1.  Global analysis of context-specific KRAS WT and mutant interactomes.**
**(A)** Number of proteins identified in KRAS WT and mutant AP-MS experiments after filtering. The conditions are unstimulated (minimal medium), DMOG (20 and 200 ng/ml in minimal medium), EGF (20 and 200 ng/ml in minimal medium), IL-6 (20 and 200 ng/ml in minimal medium), PGE2 (20 and 200 ng/ml in minimal medium), and TNF-α (20 and 200 ng/ml in minimal medium). **(B)** Overlap of all interactions identified in at least one condition with the literature KRAS interactome described in Kiel et al (2021). **(B, C)** Overlap of the same datasets as in panel (B) but focusing on effector proteins. **(D)** Principal component analysis performed on label-free quantification intensity and executed with MS log$_2$-transformed data after filtering on the whole AP-MS dataset. Colors indicate the different growth conditions, that is, DMOG, EGF, IL-6, PGE2, TNF-α, and unstimulated (Unstim), and shapes indicate the concentration of the conditions (none, 20 ng/ml, and 200 ng/ml).

To identify high-confidence interacting proteins for each condition, the label-free quantification (LFQ) intensity data for all proteins were filtered in a series of steps (see the Materials and Methods section). Specifically, data for each MS run (44 × three biological with two technical replicates = 264) were visualized as a histogram to filter out eight runs with very few proteins identified (Fig S5). Furthermore, for a protein to qualify for the high-confidence list, it had to be detected in at least 60% of the technical and biological replicates for a specific condition. In addition, only proteins that were significantly enriched compared with the beads-only control were included. Technical replicates were merged using the median LFQ intensity. To verify the robustness and applicability of the protocol, we analyzed KRAS expression levels in the complete dataset. The LFQ intensity of the KRAS bait is comparable across all AP-MS conditions (Fig S6), which suggests that a similar quantity of FLAG-KRAS proteins binds to the magnetic beads across all AP.

A total of 2,265 high-confidence PPIs were identified in 44 contexts, with an average of 725 PPIs per condition (Fig 1A). Although KRAS is a small protein, a large number of interaction partners are not too surprising, as many of those are expected to be not direct binary physical interactors but rather bind via third proteins (i.e., effectors or other proteins that enable compatible complex formation). Of note, less interactors are generally found for conditions in minimal medium (cf. unstimulated conditions in Fig 1A), supporting our initial hypothesis that microenvironmental contexts play a significant role in KRAS complex formation. Furthermore, the FLAG-KRAS WT AP generally has less PPIs, which can likely be explained by the fact that effectors bind KRAS predominantly in its GTP-bound form (in fact, no effectors are detected in any of the FLAG-KRAS WT AP grown in minimal medium). Comparing the 2,265 high-confidence PPIs determined in this work with 811 previously reported KRAS PPIs (reviewed in Kiel et al [2021]) shows an overlap of 245 proteins (hence, 30.2% of the literature PPI are among the 2,265 identified here), and 2,020 proteins were not previously reported (Fig 1B). Similar overlaps are obtained when focusing only on the classical effector proteins (Fig 1C).

To gain insights into the whole dataset, a principal component (PC) analysis and a Uniform Manifold Approximation and Projection (UMAP) (Dorrity et al, 2020) were performed with all high-confidence interactors identified in each AP-MS experiment (Figs 1D and S7). Both techniques enable a dimensionality reduction of the data and data visualization. The PC analysis, which is commonly used, tries to preserve the global structure of the data (Fig 1D), whereas the UMAP tries to preserve the data's local structure (Fig S7). The unstimulated and KRAS WT samples cluster separately from the other groups, suggesting that KRAS WT and unstimulated conditions are a good control group (KRAS mutants and stimulated conditions; middle left area in Fig 1D). Interestingly, KRAS interactor proteins detected in the different mutant datasets cluster together, compared with the different culture context datasets, where the data are more discriminated. For example, IL-6 and PGE2 contexts cluster together at the top right corner, whereas the DMOG context clusters together at the bottom right corner (Fig 1D). Taken together, these results suggest that the proteins detected in complex with KRAS seem to be more condition-dependent rather than mutation-dependent.

## Binding landscape of effectors in complex with KRAS WT and mutants

Effector proteins bind Ras in the GTP-bound state, and they are likely forming, among other proteins, the first layer of interacting proteins. Hence, we first characterized the KRAS–effector layer in more detail. As previously mentioned, no effectors are found in complex with KRAS WT in minimal medium, which is expected as KRAS will be mainly in the GDP-bound state that does not enable high-affinity binding. 11 of 56 classical Ras–effector proteins were identified in at least one of the 44 conditions (Figs 2A and B and S8). All effectors identified in complex with KRAS belong to either group 1 (AFDN, ARAF, RAF1, BRAF, and RGL2) or group 2 (RIN1, PIK3CA, GRB7, RIN2, PIK3C2A, and ARAP1) effectors. They are generally highly expressed in colon tissue and Caco-2 cells (with medium or high affinities for Ras·GTP) or are moderately expressed but have high affinities in complex with Ras·GTP (Fig 2B). Concerning the 45 effectors not found in any of the KRAS AP-MS samples, 15 belong to group 3 effectors (likely no "true" Ras–effectors) and 26 belong to group 2 (of which seven have low mRNA levels in Caco-2; <3.3 nTPM). Four effectors belong to group 1 effectors, of which RALGDS and RASSF5 are part of the KRAS literature interactome literature and are highly/moderately expressed in Caco-2 cells. SNX27 and RASSF7 are also highly/moderately expressed in Caco-2 cells, but their affinities in complex with Ras are lower. Altogether, we provide a near-to-complete binding landscape of effectors in complex with KRAS under the conditions tested.

A comparison with the effectors identified in previous KRAS interactomes (reviewed in Kiel et al [2021] (Fig 2C and D) shows that the AP-MS experiments in this study specifically increase the percentage of group 2 effectors, but little increase in group 1 effector coverage and no increase in group 3 effectors (Fig 2E). Furthermore, the number of conditions in which an effector is identified in complex with KRAS tends to be lower for group 2 effectors (Fig 2F). To visualize in which genetic and culture contexts the 11 effectors were detected, two heatmap images were generated (Fig 3). The two heatmaps show a similar pattern in terms of effector detection and abundance in each of the groups of the AP KRAS mutants (i.e., G12D, G12C, and G12V) in unstimulated and stimulated conditions. More specifically, the effectors AFDN, ARAF, RAF1, and RIN1 are detected in all the KRAS mutant AP-MS experiments when unstimulated and stimulated (Fig 3A). These effectors are also detected in the WT AP-MS experiments—albeit not in all the culture contexts. They are classified in the effector group 1 except for RIN1, which is part of the effector group 2. Moreover, they appear to be more abundant when detected in particular conditions such as DMOG and TNF-α, compared with other conditions such as IL-6 or PGE2 (Fig 3B). As these effectors are detected consistently in the presence of KRAS with or without stimulations, we propose that these effectors are KRAS-specific rather than condition(stimulation)-specific. Other effectors are only detected in specific stimulated conditions and are mainly detected in the predicted effector group 2. Moreover, GRB7 is only detected in the presence of the DMOG culture context. Another effector, PIK3CA, is only detected when stimulated with TNF-α in the presence of either the G12D or the G12V KRAS mutations. The effector PIK3C2A is only detected significantly in the presence

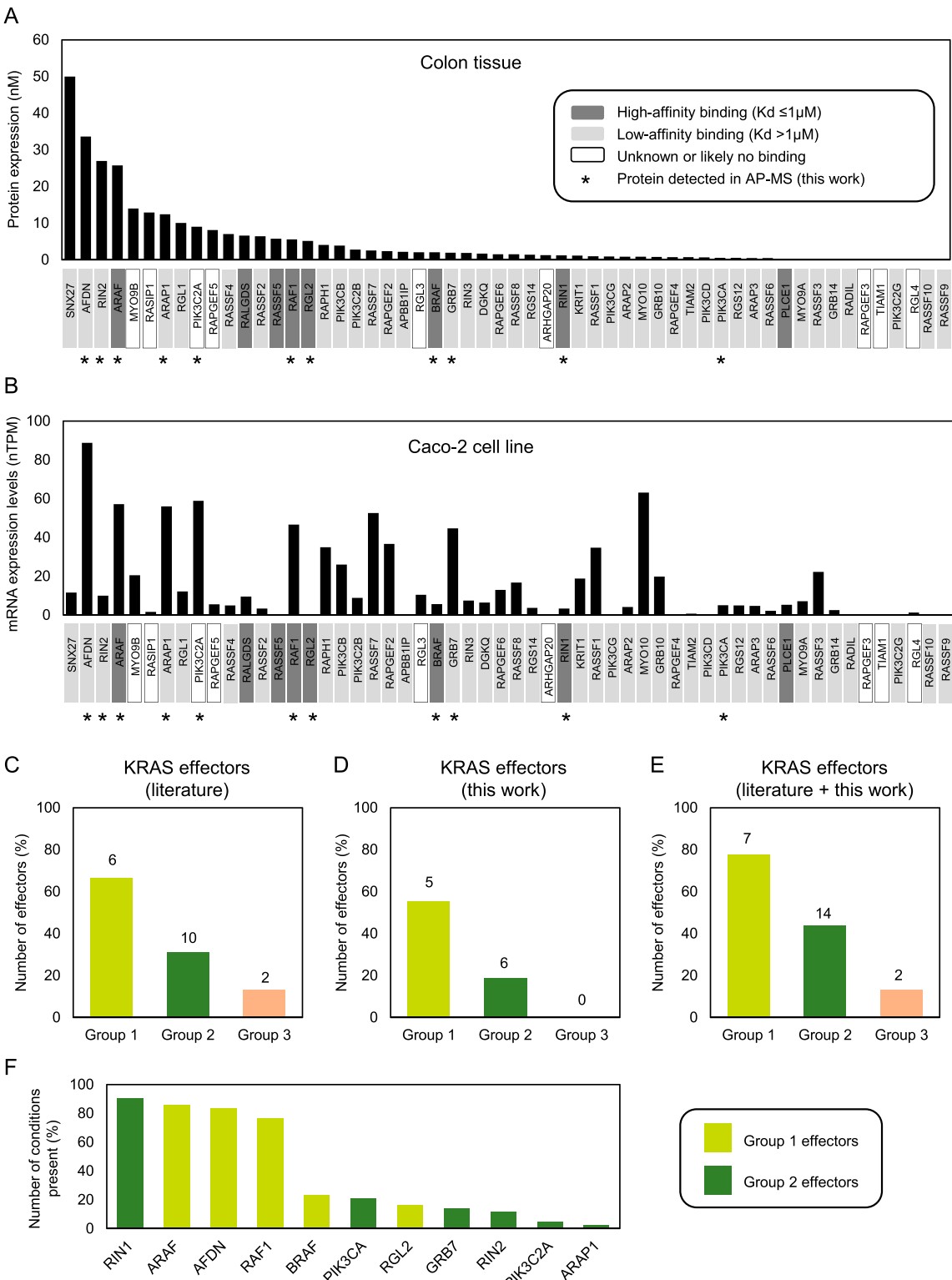

**Figure 2. Ras–effector abundance, binding affinities, and detection in AP-MS experiments.**
**(A, B)** Ras–effector protein abundance in human colon tissue based on Gimple and Wang (2019) and Ibáñez Gaspar et al (2021) (A) and mRNA abundance in Caco-2 cells based on the Human Protein Atlas database (B). The colors of the effectors' names at the bottom of each histogram correspond to Ras–effector binding affinities. The black star indicates effectors that were detected in at least one of the AP-MS experiments conducted in this work. TPM, transcripts per million; nTPM, normalized transcript expression values per sample; Kd, dissociation constant. **(C, D, E)** Effectors identified in at least one condition in the literature KRAS interactome described in Kiel et al (2021) (panel (C)), in this work (panel (D)), and in the combined datasets (panel (E)). Group 1, group 2, and group 3 effectors are normalized based on the total number of effectors in the respective group. **(F)** Number of conditions where an effector is present.

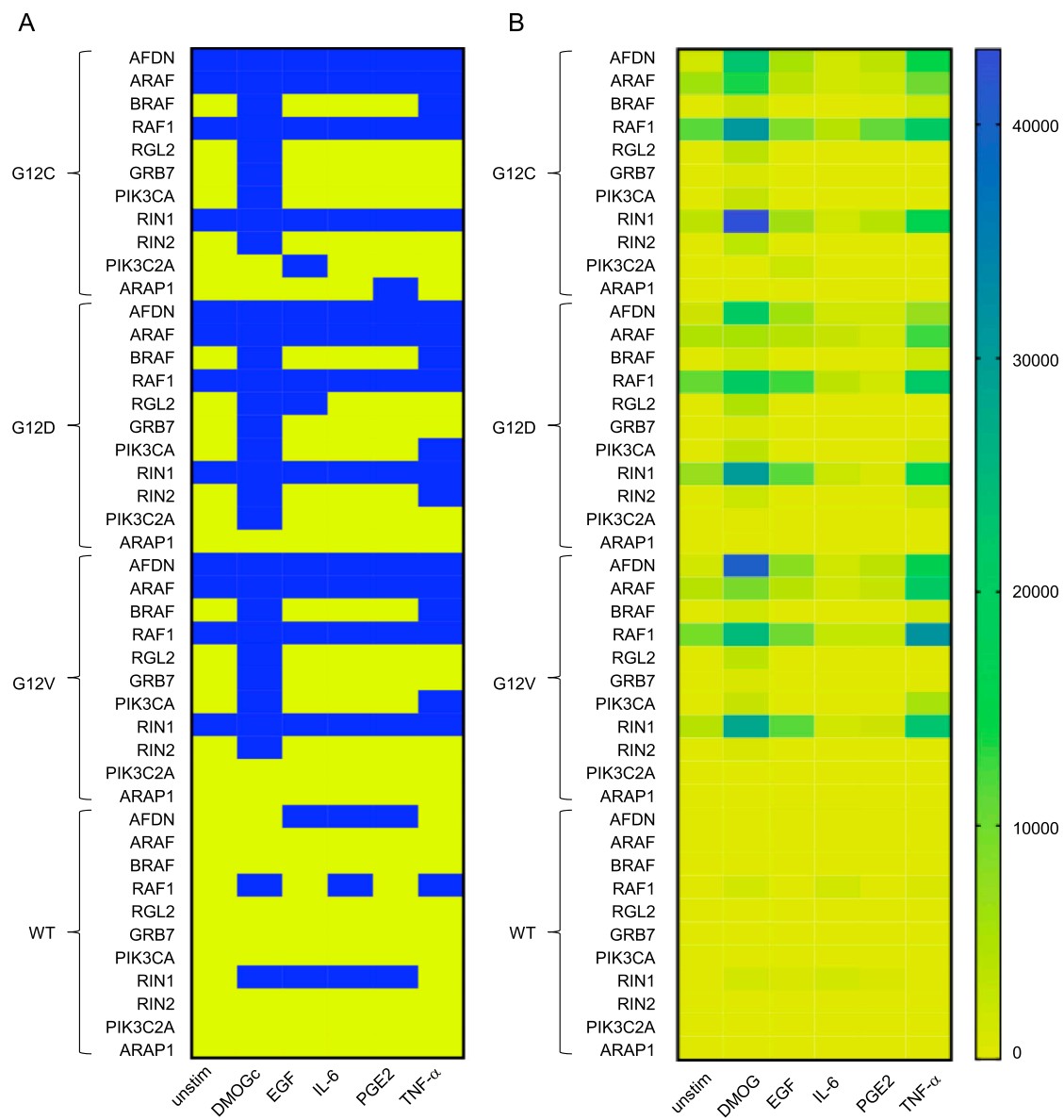

**Figure 3. Summary of effector presence in AP-MS experiments in Caco-2 cells.**
The rows display the effectors in complex with KRAS grouped by mutational status (WT, G12D, G12V, and G12C), and the columns represent the conditions (unstimulated or stimulated with either DMOG, EGF, IL-6, TNF-α, or PGE2). **(A, B)** To generate the two heatmaps, the LFQ intensities were analyzed, the different concentrations for each stimulation were merged and translated for the heatmap (A) in terms of the presence or absence of an effector (color code: detected = blue and not detected = yellow), and for second heatmap (B), the LFQ intensities were directly plotted into the heatmap (code: from low abundance = yellow to high abundance = blue). The heatmaps were created using GraphPad Prism 9.

of the G12D KRAS mutation with DMOG. These effectors grouped in effector group 2 can be classified as conditions-specific. To mention, the effector BRAF, which was computationally predicted to be always in complex with KRAS, is found in complex with KRAS only in DMOG and TNF-α conditions.

Altogether, this supports our initial hypothesis that group 2 effectors tend to be found in complex with Ras only in specific conditions that promote PM recruitment via RBD-independent mechanisms (Fig S9). Furthermore, it validates our computational-based classification into group 1, group 2, and group 3 effectors

(Catozzi et al, 2021) and supports its applicability beyond the 29 human tissues as the basis of the prediction model.

## Investigation of functional differences in the KRAS interactome

To investigate functional differences in the interactome of the different genetic and culture contexts, two approaches were chosen. First, a differential interaction analysis was performed on the identified proteins followed by a gene set enrichment analysis against the gene ontology (GO) biological processes (Fig 4A).

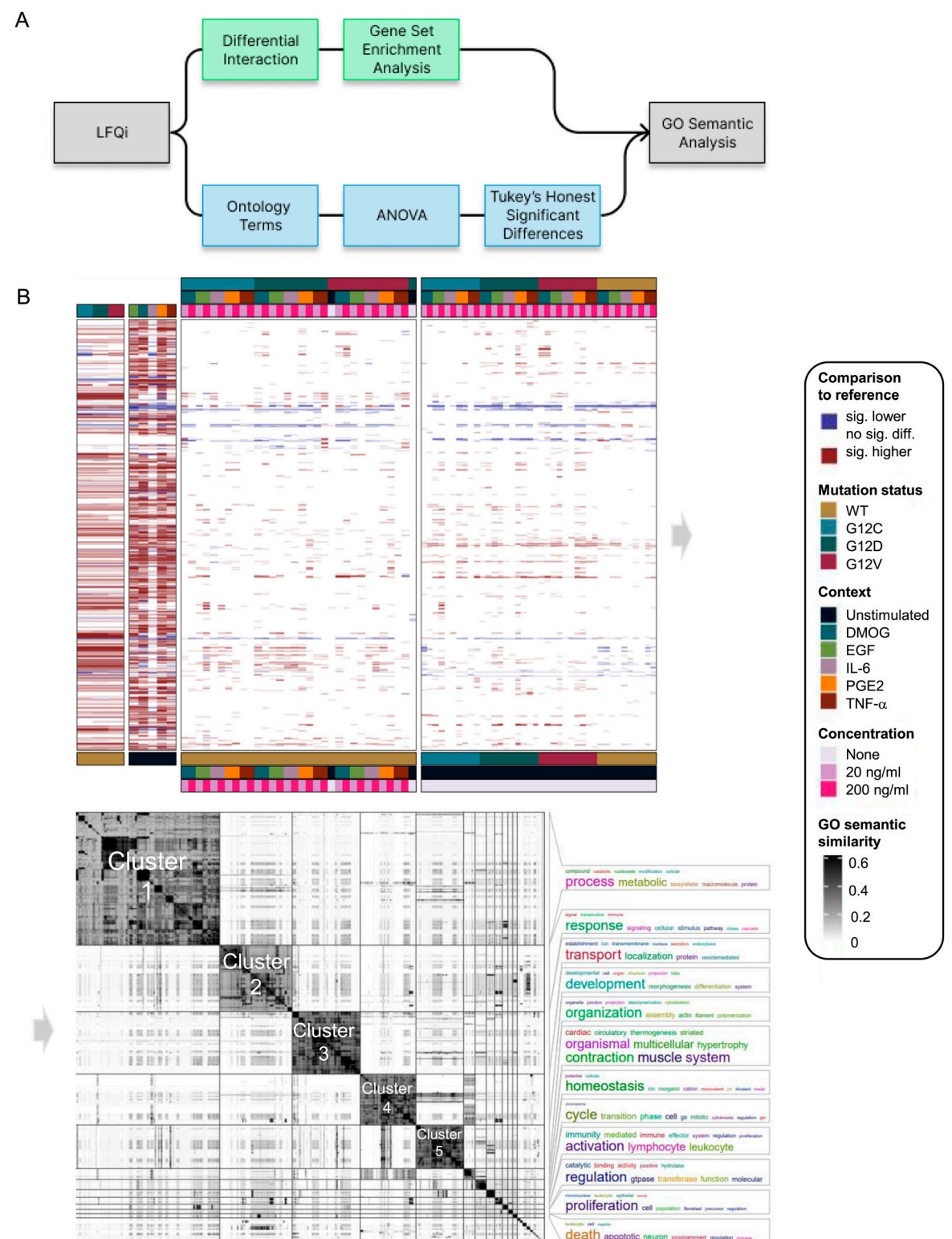

**Figure 4. Gene set enrichment analysis (GSEA) of AP-MS analysis in different genetic and culture contexts.**
**(A)** Overview of the functional analysis pipeline. The first approach is a standard differential analysis pipeline using limma, followed by a gene set enrichment analysis against the GO "biological process" terms. The second approach is to sum up LFQ intensities for each of the GO "biological process" terms. Then, for each term, an ANOVA was performed to identify whether there was a significant influence of mutation status, condition and concentration, and their interaction terms. Both analyses were then analyzed using semantic analysis of ontology terms. **(B)** Distance matrix with the pairwise semantic distances between the GO terms is shown in the center of the plot. On the right, the bigger clusters are annotated by word clouds. On the left, the data from the ANOVA and the GSEA pipeline are shown, for the ANOVA, whether a main effect is statistically significant; and for the GSEA, whether a significant enrichment was found in the relevant pairwise comparison. All data shown are in comparison with the WT and unstimulated for the analysis of mutation status and condition, respectively.

 **Life Science Alliance**

Second, using the ontology, we collapsed each sample onto all ontologies listed under "biological process" by summing up their LFQ intensities. This process preserved the variation in the data (Fig S10). Then, multiple ANOVAs were used to identify differences between the samples in terms of their summed-up LFQ intensities. Of the 16,000 GO terms tested, we find significant changes for 2,135 (Fig 4B and S11A). Afterward, semantic analysis was used to organize the significantly changed ontology terms into clusters. The full distance heatmap from the semantic analysis together with the clusters and some of the data projected onto it is shown in Fig 4B. The biggest semantic clusters are linked to metabolic and biosynthetic processes (cluster 1), signaling and immune response (cluster 2), vesicle-mediated transmembrane and ion transport (cluster 3), differentiation, development, and morphogenesis (cluster 4), and actin and cytoskeleton organization (cluster 5) (Figs 4B and S11B–H). Smaller clusters are linked to thermogenesis, ion homeostasis, cell cycle, leukocyte activation, regulation of GTPases, proliferation, and apoptotic cell death (Fig 4B). Altogether, the overall functional differences in the KRAS interactome are consistent with known cellular functions mediated by KRAS (Ibáñez Gaspar et al, 2021; Catozzi et al, 2022).

The above twofold analysis shows different aspects of functional differences between the different genetic and culture conditions. To make the data better approachable, we developed an interactive R Shiny app for exploring the functional terms that are different between the samples (Fig S12). Users can explore the analysis through the semantic distance heatmap and the semantic clusters, or search and filter for functional terms of their interest, visualize which proteins are part of this particular GO term, and show their abundance in the different samples. It also directly displays the samples that are statistically significant to each other. We propose this app as a resource to filter for interesting functional influence of certain KRAS mutations or certain growth conditions from our dataset. From the results of this functional investigation, we selected some GO terms of interest to us, which we went on to validate in the wet laboratory. Those were GO terms related to proliferation ("Epithelial cell proliferation" and "Positive regulation of cell population proliferation"), glucose metabolism ("Glycolytic process" and "Regulation of glucose metabolic process"), and ATP metabolism ("ATP metabolic process" and "Regulation of ATP metabolic process") shown for genetic contexts G12C and G12D and culture contexts DMOG and IL-6 in Fig 5A. As we are studying the effect of oncogenic KRAS mutation in an adenocarcinoma cell line under CRC microenvironment mimicking culture contexts, we were particularly interested in biological processes commonly observed in cancer development. Among the hallmarks of cancer (Hanahan & Weinberg, 2011), sustained cell proliferation and the Warburg effect, described by a switch in cell energetic metabolism from oxidative phosphorylation to aerobic glycolysis, are two biological and metabolic processes that are feasible to test experimentally. In the GO functional analysis, we observed that PGE2 and DMOG had the greatest number of differentially expressed GO terms compared with unstimulated, whereas IL-6 had the lowest. In addition, we observed that the differential expression profile of the three KRAS mutants was similar, and

KRAS G12C and G12D differ the most for GO terms related to biological and metabolic processes.

## Analysis of cell phenotypes in selected genetic and culture contexts

To investigate changes in cell proliferation and glycolytic metabolism for three genetic contexts (KRAS WT, G12D, and G12C) in three culture contexts (unstimulated, DMOG, and IL-6), we measured cell count, cell viability, glucose uptake, and lactate release over a 72-h period in Caco-2 cells (see the Materials and Methods section) (Fig S14A–H). We chose to determine Caco-2 metabolism and proliferation until 72 h post-transfection under the assumption that the changes observed in the interactome seen 24 h post-transfection would affect cell phenotype on a long-term basis and that phenotypic changes can take longer to be accurately measured. To ensure that exogenous KRAS was expressed until 72 h post-transfection, we performed a Western blot analysis of FLAG-KRAS before transfection and from 24 h to 72 h post-transfection. We observed an overall significant increase in FLAG expression 24 h post-transfection with a peak at 48 h, and then a decrease in FLAG expression at 72 h to reach the level of 24-h time point (Fig S13). Those results suggest that for all genetic contexts, exogenous KRAS was effectively expressed during a 72-h period. However, we also noticed that exogenous KRAS expression was significantly different between the three genetic contexts.

We observed a significant increase in cell proliferation under the DMOG context compared with the control (unstimulated), although IL-6 led to a significant decrease in cell proliferation (Fig S14A). The increase in cell proliferation in the DMOG culture context was confirmed by cell viability results (ATP concentration; Fig S14G). Furthermore, for those two phenotypic parameters, under DMOG stimulation, KRAS G12D had a significantly greater cell proliferation than KRAS WT and G12C. In terms of glycolytic metabolism, it was overall better captured by lactate release than by glucose uptake and the highest lactate release was observed in the DMOG context. A greater lactate release was also observed for both KRAS mutants compared with WT (Fig S14E). When normalized per cell, both glucose uptake and lactate release were significantly greater in the IL-6 condition than in the control and DMOG, suggesting a higher glucose use at the single-cell level in the IL-6 context. Finally, the ATP pool per cell, which can be used as a proxy for changes in metabolic activity, was higher for both stimulations compared with the control and for both oncogenic KRAS compared with the WT. Altogether, the experimental results suggest that under genetic and culture contexts that mimic colon and CRC, Caco-2 cells produce more energy than KRAS WT cells in control conditions. Hence, the bioinformatic analysis of KRAS complex composition correlates with the phenotypic changes.

Next, we compared the measured phenotypic parameters with the sum of LFQ intensities of proteins associated with GO terms linked to proliferation, glucose metabolism, or ATP metabolism with each term directly related to the final cell phenotype or its regulation (Fig 5B and C). Overall, the culture context effects are similar between the AP-MS (GO terms) and the phenotypic experiments. Indeed, we observed that for all GO terms except "Glycolytic process," the sum of LFQ intensities was significantly higher in DMOG

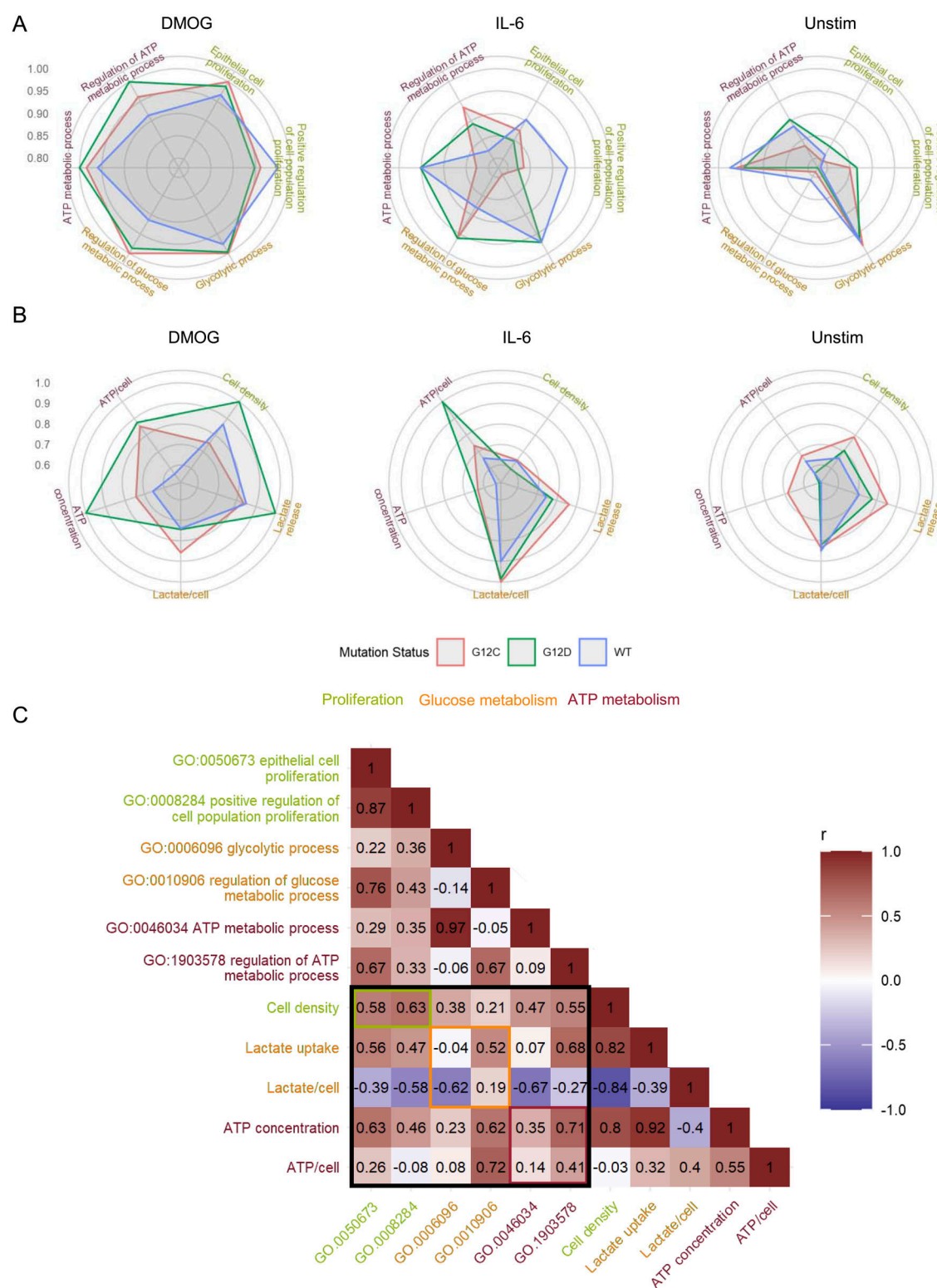

**Figure 5. Comparison of phenotypic parameters with the sum of LFQ intensity of related GO terms in the AP-MS.**
**(A)** Radar plot of phenotypic parameters related to cell proliferation, glucose metabolism, and ATP metabolism of Caco-2 cells under different genetic and culture contexts. **(B)** Radar plot of the GO terms related to cell proliferation, glucose metabolism, and ATP metabolism of Caco-2 cells under different "culture" and "genetic" contexts. Values in radar plots are the LS mean values normalized by the maximum value of each parameter to obtain values between 0 and 1. **(C)** Pearson's correlation matrix of phenotypic parameters versus sum of LFQ intensity of specific GO terms from the AP-MS. Phenotypic parameters and GO terms related to the same biological process are displayed with the same colors.

compared with unstimulated and IL-6 (Table S1). Those results are similar to the cell density, lactate release, and ATP concentration experimental results. In addition, for "Regulation of glucose metabolic process" the sum of LFQ intensity was higher in the IL-6 context compared with unstimulated, which could be linked to the higher lactate release per cell observed in the IL-6 condition. The only notable difference is observed for the "Glycolytic process" where the sum of LFQ intensities was significantly lower in the IL-6 culture context, which is contrary to what was observed for lactate release per cell. Results of the effect of the genetic context are also to some extent similar, with no significant effect on proliferation-related GO terms such as cell density. In addition, a significantly higher sum of LFQ intensities was observed in the "Regulation of glucose metabolic process" for the two KRAS mutant proteins compared with WT, with results similar to the lactate release. However, no genetic context effect was observed for ATP metabolism–related GO terms, although mutant KRAS had a higher overall ATP concentration than WT. Particularly, the sum of LFQ intensities of "Glycolytic process" was significantly lower in G12C than in the two other genetic contexts. This was mostly due to a substantial decrease under IL-6 culture conditions, which is opposite to what was found for overall lactate release. Those results suggest that the strongest changes in phenotype, such as those observed for DMOG stimulation, are more reliably captured by the AP-MS than the milder changes such as those observed for the genetic context or IL-6 stimulation. Furthermore, the changes observed for GO terms referring to the regulation of a process seem to be better at capturing the results of phenotypic experiments than GO terms referring to the biological process itself. The latter point is affirmed by better Pearson's correlation coefficient observed for phenotypic parameters with the associated GO terms referring to the regulation of a biological process than those directly referring to the biological process (Fig 5C). This may be because proteins that have a direct effect on the phenotype are too many layers downstream of the Ras signaling network to be captured by the AP-MS experiments. Finally, for metabolic parameters, the functional analysis results are more correlated with overall lactate release and ATP concentration than with the values normalized per cell. This implies that the results of the functional analysis based on AP-MS data are more suitable for capturing changes in metabolism at the level of the whole-cell culture than at the level of a single cell.

### Information flow analysis predicts the contribution of effectors to functional processes

After demonstrating that the differences in the summed LFQ intensities for a functional process can be associated with functional differences in the Caco-2 cells, we aimed to further explore how the changes in the interactome led to these functional differences. In particular, we were interested in understanding which effectors and proteins downstream of KRAS were involved in a specific process. To this end, we used random walks over a filtered version of the STRING network, in which we biased the random decision for each step depending on whether a potential next protein was part of the AP-MS dataset for a specific sample or not (see the Materials and Methods section). This left us with a collection of paths and their

probability to be traversed, based on the network architecture and the proteins found in the AP-MS sample.

Applying this method to the functional terms and samples of interest, we observed pronounced changes in the network architecture depending on the sample (Fig 6). Particularly, for some GO terms there was a different predicted engagement of effectors. Examples are the two GO terms related to "Epithelial Proliferation" for KRAS oncogenic mutations in culture contexts IL-6, DMOG, and unstimulated (Fig S15). For "Epithelial cell proliferation," in the IL-6 context high path counts are found for the effectors AFDN, ARAF, and RAF1 (Fig 6A). In the DMOG context, additional high counts are found for BRAF, GRB7, and PIK3CA (Fig 6A). Likewise, for "Positive regulation of cell population proliferation" contributions of AFDN and PIK3CA dominate in the DMOG context (Fig 6B). For GO terms related to glucose metabolism, AFDN, GRB7, and PIK3CA dominate the path counts for "Glycolytic process (GO:0006006)" in the DMOG condition and for KRAS G12C in the IL-6 context, but have a low count for "Regulation of glucose metabolic process (GO:0010906)," where ARAF, BRAF, and RAF1 dominate in most culture conditions (Figs 6C and D and S16). With respect to GO terms related to ATP metabolism, the culture contexts IL-6 and DMOG show profound path count differences for the effector RAF1, which contributes more in the DMOG than in the IL-6 context (for the KRAS G12D genetic context), but slightly higher path count in the IL-6 context for KRAS G12C (Figs 6E and F and S17). The path count is generally low for BRAF and ARAF. AFDN has high/the highest path counts in almost all genetic and culture contexts. Also noteworthy, RIN1 is present in all contexts but not necessarily much involved. Altogether, our biased random walk analyses predict the contribution of individual effectors to GO terms that link to experimental phenotypes.

## Discussion

This study set out to explore KRAS as a key cellular signaling hub in specific relevant (patho)physiological contexts. The Caco-2 cell line has been used as a relevant model system that can be grown in various growth media (culture contexts) and enables the exogenous expression of KRAS WT and oncogenic mutants (genetic contexts). Indeed, Caco-2 cells are human intestinal epithelial cells that closely mimic the colon intestinal epithelium in the early stage of CRC. By identifying different levels of network organization (e.g., subcomplexes and number of paths traversing a network), we aimed to detail the downstream pathway of KRAS further and investigate the functional outputs. To address this challenge experimentally, even though all methodology has their limitations, AP-MS excels in profiling interactomes in humans (Hein et al, 2015) because of its sensitivity and its ability to detect interactions within complexes in appropriate contexts (Huttlin et al, 2021). We successfully pulled down complexes using the exogenous expression of tagged bait proteins for different KRAS variants in Caco-2 cells.

Effectors bind to Ras in a mutually exclusive fashion and can potentially compete for binding (Kiel et al, 2013). Our earlier computational predictions suggested that there is a considerable impact of culture contexts on the recruitment of specific effectors

**Figure 6.   Pathway analysis by biased random walks.**
**(A)** Heatmaps of effector traversal in the different genetic and culture contexts for selected GO terms associated with cell phenotypes. Epithelial cell proliferation (GO: 0050673). **(B)** Positive regulation of cell population proliferation (GO:0008284). **(C)** Glycolytic process (GO:0006096). **(D)** Regulation of glucose metabolic process (GO: 0010906). **(E)** ATP metabolic process (GO:0046034). **(F)** Regulation of ATP metabolic process (GO:1903578). The color scale indicates how often an effector is traversed.

to the PM (Catozzi et al, 2021). Here, we identified a total of 11 effectors in at least one of the AP-MS experiments, of which seven are only found in complex with KRAS in some genetic and culture contexts. For example, effectors such as PI3KCA, RIN2, GRB7, or ARAP1 are detected in the presence of culture conditions such as hypoxia (HIF stabilization), IL-6, and TNF-α. We predict that in these cases, the affinity between the RBD and KRAS is not high enough to allow for sufficient binding and that additional domains present in effectors are required to increase the number of complexes formed between KRAS and effectors at the PM (based on the "piggyback" mechanisms [Kholodenko et al, 2000]). Indeed, we show in this work that the total number of effectors and other proteins in the Ras-mediated complex increases with the number of conditions. This context-dependent binding can be explained by the fact that cells in their physiological microenvironment are constantly experiencing a variety of stimuli that trigger receptors (e.g., the EGF receptor) located on the PM, where Ras is located (Eisenberg & Henis, 2008). In the context of cancer, tumor cells are often located in a hypoxic, immunosuppressive, and nutrition-deficient microenvironment that causes reprogramming of metabolism and signaling (Hanahan & Weinberg, 2011). Indeed, we identified culture context–specific metabolic alterations in glucose and ATP metabolism in the Caco-2 cells. Hence, this work supports the requirement to study the role of the microenvironment when performing an experiment that aims to characterize PPI networks because they have a major role in rewiring complex formation. It also demonstrates the need to consider multidomain interactions. However, the interpretation of PM recruitment and culture conditions might not be straightforward. In fact, it is difficult to understand and predict what happens on the upstream level of Ras–effector interactions and why some effectors are identified in the specific conditions tested. This is partly due to signaling pathways that are highly cell type–specific (van Boxel-Dezaire et al, 2006; Miller-Jensen et al, 2007). Together with initiatives such as the Human Protein Atlas (Uhlén et al, 2015) and the large-scale interactome (AP-MS–based) BioPlex database (Huttlin et al, 2015, 2017), analyzing human PPI and the conditions in which they occur will be essential for the creation of a context-dependent human interactome knowledge base. Toward reaching this goal, a new version of the BioPlex 3.0 interactome has been recently published where, in addition to HEK293T cells, a dual comparison with the HCT116 cell line was performed (Huttlin et al, 2021).

Based on the assumption that effectors compete for binding to KRAS, our working hypothesis is that individual KRAS/effector-mediated subcomplexes form in a cell, which ultimately affect downstream signal propagation and cellular phenotypes. Indeed, we show here that differences in KRAS-mediated complexes propagate to downstream changes in phenotypes that roughly align with the predicted functional changes based on GO terms of proteins detected in an AP-MS experiment. This suggests that the PPI network orientation/assembly on the level of KRAS (likely mediated in part by effectors in complex with KRAS) impacts the downstream phenotype.

The analysis of PPI in mammalian cells is challenging, and different methods have both advantages and limitations (Snider et al, 2015). In the experimental AP-MS setup used in this work, FLAG-KRAS WT and mutant variants are exogenously expressed. A

main advantage of AP-MS is that it can be performed in a high-throughput fashion and that epitope tagging allows the study of proteins for which antibodies are not available or not suitable for immunoprecipitation. A limitation of AP-MS is that it does not allow for the detection of spatial or temporal PPI because of the need to perform cell lysis and AP. Other techniques, such as proximity ligation assays (PLA), can determine PPI in a spatial and temporal manner, but require high-quality antibodies and do not have high-throughput capability (Snider et al, 2015). Indeed, when we performed PLA on non-transfected Caco-2 cells, we found increased KRAS/PM colocalization of the effector PI3K in the EGF culture context that changes over time (Fig S18). This highlights a central question in biology as to how changes (often short term) in cellular protein complexes result in phenotypic changes that manifest over longer time periods (Rukhlenko et al, 2022).

As effectors compete for binding to KRAS, we hypothesized that specific Ras–effector subcomplexes exist that each (or in combination) links to specific phenotypes. To explore the contribution of individual effectors to phenotypes, we used biased random walk analysis. Indeed, we find differences in the number of paths between different genetic and culture contexts. The analysis also enabled us to predict which effector pathways are likely linked to cellular phenotypes. Hence, our analysis pipeline that combines AP-MS data with random walks and GO terms provides a novel way to link PPI networks to phenotypes. The pipeline and code are available to the scientific community and can be adapted for specific AP-MS experiments. There are, however, limitations of the random walk analysis. The data structure at the end is a collection of different paths for different targets for different conditions. Some of these are comparable, and some are biased. Paths ending into the same target should be comparable across conditions, as long as the underlying network structure does not change. However, there is a bias for shorter paths to be more likely to be found, and targets with shorter shortest paths have on average higher counts in the found paths. In addition, the analysis is strongly dependent on the underlying network structure that is used.

With respect to the impact of genetic versus culture contexts on KRAS-mediated network rewiring, our analyses based on PCA and UMAP suggest that the impact of growth condition (culture context) is greater than the type of oncogenic mutation (genetic context). We observe a similar trend in the functional analysis and in the effector contributions as calculated by random walks, where different oncogenic mutants generally having more similar effector path counts are different culture contexts/growth conditions. Indeed, the results of this work offer an additional explanation why cancer genes and mutations only manifest in some but not all tissues (Schaefer & Serrano, 2016).

Future steps in systems medicine require the integration of protein abundance with context-specific conditions and localized signaling responses. Indeed, quantitatively predicting the influence of specific conditions on larger networks to get an efficient predictive model would be ideal, especially in the case of oncogenic mutations. In addition, understanding the rewiring in physiological contexts to enhance the understanding of network rewiring in cancer contexts would provide new insights into potential therapeutic targets (Nogales et al, 2022).

# Materials and Methods

## Culturing of Caco-2 cells

Caco-2 cells (ATCCHTB-37) were cultured in DMEM (21969-035; Gibco, Thermo Fisher Scientific) supplemented with 2 mM l-glutamine (25030-024; Gibco, Thermo Fisher Scientific), 10% (vol/vol) FBS (A4766801; Gibco, Thermo Fisher Scientific), and 1% penicillin/ streptomycin (15140122; Gibco, Thermo Fisher Scientific). For long-term storage, frozen stock vials were made on the week of receiving the cell line in Recovery cell culture freezing medium (12648010; Gibco, Thermo Fisher Scientific) and stored in liquid nitrogen. For each experiment, cells were not exceeding passage 25 and were thawed from the liquid nitrogen stock. To generate growth media that mimic conditions relevant in the colon and CRC ("culture contexts"), the minimal medium (DMEM with 2 mM l-glutamine) was supplemented with either IL-6 (interleukin-6) (Thermo Fisher Scientific), TNF-α (Thermo Fisher Scientific), PGE2 (Thermo Fisher Scientific), EGF, or the HIF-hydroxylase inhibitor DMOG (Cayman Chemical) at different concentrations (20 and 200 ng/ml). Caco-2 cells for PLA were kindly gifted by Professor Per Artursson.

## Plasmids for exogenous expression of FLAG-KRAS WT and mutants

Plasmids were gifted from the previous research laboratory of Christina Kiel in Barcelona (CRG) (from Luis Serrano and Hannah Benisty). All the plasmids harbor the identical backbone pMDS-TetOn3G-kozak-FLAG-GOI (gene of interest). Plasmids differ only by their GOI, which are WT KRAS, KRASG12D, KRASG12V, or KRASG12C as GOI.

## Bacterial transformation with plasmids, plasmid extraction, and purification

The bacterial transformation of the plasmids for exogenous expression of FLAG-KRAS (pMDS-TetOn3G-kozak-FLAG-KRAS WT/mutants) was performed using the One-Shot Stbl3 (C737303; Invitrogen) chemically competent bacterial cells to replicate each plasmid following the manufacturer's instructions. Subsequently, 100 μl of the bacteria–plasmid solutions was plated into LB selective agar plates containing the antibiotic spectinomycin (50 μg/ml). Plates were incubated at 37°C, overnight. The next day, individual bacterial colonies were selected from a LB agar plate and grown in 4 ml LB broth with the corresponding antibiotics for 6–12 h at 37°C in a shaker-incubator at 250 rpm. After incubation, several aliquots of this original starter culture were used to generate a bacterial glycerol stock for long-term storage at –80°C (1 ml transformed bacteria in 1 ml 50% glycerol). The remainder of the original starter culture was then used to grow at a large scale the transformed bacteria under selective antibiotics overnight in 500 ml of LB medium at 37°C in a shaker-incubator at 250 rpm. The HiSpeed Plasmid Maxi Kit (QIAGEN) was used to generate a larger amount of the FLAG-KRAS plasmids. The kit was used following the manufacturer's instructions. The final DNA was eluted in 400 μl of TE buffer and allowed to resuspend overnight to ensure homogeneity. The

next day, concentrations and purities were measured on the Implen NanoPhotometer NP80, and plasmid DNA was stored at –20°C.

## Transfection and expression of FLAG-KRAS in Caco-2 cells

For AP-MS experiments, Caco-2 cells were seeded 24 h before transfection in 10-cm dishes in normal growth medium and grown to 70–80% of confluency. Cells were transfected with 15 μg of pMDS-TetOn3G-kozak-FLAG-GOI plasmids (containing FLAG-KRASWT or FLAG-KRASG12D or FLAG-KRASG12V or FLAG-KRASG12C as GOI) using Lipofectamine 2000 (11668-019; Invitrogen) according to the manufacturer's instructions in Opti-MEM reduced serum medium (31985-062; Gibco, Thermo Fisher Scientific) for 4 h. Then, the medium was changed and supplemented with culture medium containing the various growth conditions. To note, cells transfected with the KRASWT plasmids were always supplemented with 15 ng/ml of doxycycline (Sigma-Aldrich). Cells were incubated for 24 h at 37°C and harvested. FLAG-KRAS mutant plasmid transfections were not supplemented with doxycycline as the promoter is leaky and KRAS mutant proteins were already expressed at WT levels without adding doxycycline.

## Caco-2 cell lysis, protein extractions, and concentration

Caco-2 cell lysates were obtained after trypsinization, and cell pellets were recovered and washed twice with PBS 1X. The cell pellets were then resuspended in the appropriate volume (e.g., 300 μl for the AP-MS experiments) of lysis buffer (50 mM Tris–HCl, pH 7.5, 1 mM EDTA, 1 mM EGTA, 150 mM NaCl, 2 mM MgCl2, 1 mM DTT, and 1% IGEPAL/NP-40 supplemented with PhosSTOP [Roche] and cOmplete, Mini protease inhibitor cocktail [Roche]). Cells were lysed for 30 min on a rotator at 4°C and centrifuged at 14,000 rpm for 30 min at 4°C, and the supernatants were collected in a new tube. Protein concentrations were measured using the Pierce 660-nm Protein Assay (22660; Thermo Fisher Scientific) per the manufacturer's guidelines. Samples were incubated for 5 min before absorbance was read at 660 nm on a SpectraMax M3 plate reader. Net absorbance values were plotted against BSA protein concentration for standard curve generation (23208; Thermo Fisher Scientific). For each sample, the concentration was obtained by comparing net absorbance values against the generated standard curve. A new standard curve was generated for each assay. Kept on ice, cell lysates were then directly used for AP.

## Western blotting

Before loading the samples into the gel, a normalization of the concentration for each sample is done, with a concentration aiming to be 1 μg/μl. Proteins were then denatured by incubating samples at 95°C for 5 min in 4 × Laemmli buffer and DTT before loading onto 4–12% NuPAGE gradient precast gels (Thermo Fisher Scientific). Gels were run for 10 min at 110 V, followed by 45 min at 150 V, with gels submerged in NuPAGE MES running buffer (Thermo Fisher Scientific). After electrophoresis, proteins are dry-transferred using the iBLOT2 device (Thermo Fisher Scientific) for 7 min into a nitrocellulose membrane. The membranes were checked by Ponceau S staining to ensure protein transfer. Then, the membranes were washed in 1 X Tris buffer saline/Tween-20 (TBS-T) before blocking solution for 1 h in 5% milk at room temperature in a shaking device.

Depending on the antibody, the membranes were incubated either overnight at 4°C or at room temperature for 4 h, with the primary antibody diluted in 0.05% milk in TBS-T. The membranes were then washed three times in TBS-T, 10 min each, and incubated with HRP-conjugated secondary antibody for 1 h diluted in milk TBS-T on a shaking device. Protein bands were developed using a high-sensitivity ECL reagent (Thermo Fisher Scientific) with the West Pico Western blotting substrate per the manufacturer's instructions and visualized using the G-Box image developer (SYNGENE). Densitometry analysis was performed using ImageJ, with target protein bands normalized to a loading control (β-actin or GAPDH). The following antibodies were used for Western blotting: β-actin (#4970, rabbit/monoclonal, 1/3,000 dilution; Cell Signaling), GAPDH (ab2118, rabbit/monoclonal, 1/1,000 dilution; Abcam), pan-Ras (ab52939, rabbit/monoclonal, 1/5,000 dilution; Abcam), KRAS (CPTC-KRAS4B-2, DSHB, mouse/monoclonal, 0.5 μg/ml working concentration), and secondary anti-mouse HRP (ab97023, goat/monoclonal, 1/3,000; Abcam).

## AP

Caco-2 cell lysates expressing FLAG-KRAS proteins were immunoprecipitated from 800 μg of cell lysate using anti-FLAG-M2 magnetic beads (M8823; Sigma-Aldrich) using the KingFisher DuoPrime purification system (Thermo Fisher Scientific). Beads were washed in TBS (according to the manufacturer's instructions) for 5 min, twice, at low speed. Then, beads were collected by the KingFisher magnet and discarded into the samples wells and mixed at a slow speed for 1 h. Beads–antibody–samples were collected and went through different wash salted solutions (Wash 1 and Wash 2: RIPA buffer with 150 mM NaCl; Wash 3: RIPA buffer with 500 mM NaCl), mixed at low speed for 30 s. Beads–antibody–samples were eluted in 50 μl of glycine (0.1 M, pH 3.0) for 5 min. Immediately after, samples were neutralized with 20 μl of Tris base (1 M, pH 8.0).

### Caco-2 cell proteome in different culture contexts

Caco-2 cells were seeded in 6-cm dishes at $8 \times 10^5$ cell/dish (about 70% confluency) in 4 ml normal growth medium. 24 h post-seeding, cells were transfected with 5 μg of pMDS-TetOn3G-kozak-FLAG-GOI plasmids (FLAG-KRASG12D) or TE buffer alone (non-transfected control) using Lipofectamine 2000 (11668-019; Invitrogen) in serum-free DMEM for 4 h. Then, the medium was changed and replaced by medium containing the various growth conditions at 20 ng/ml (unstimulated, DMOG, IL-6, PGE2, EGF, and TNF-α). After 18 h, cells were harvested and directly lysed to perform both Western blot and MS sample preparation (without AP).

### Sample preparation after AP for MS

For protein cleanup, the paramagnetic bead–based SP3 (solid-phase–enhanced sample preparation) workflow was used (Hughes et al, 2019). For each AP experiment, sample protein concentrations were determined using the Pierce BCA protein assay (Thermo Fisher Scientific) following the manufacturer's instructions, and 50 μg of proteins was adjusted in 20 μl of buffer/MS-grade water. Samples were homogenized and denatured in urea (final concentration, 4 M), ammonium bicarbonate (100 mM), and calcium chloride

(100 mM), then reduced in DTT (final concentration, 1 mM) for 15 min at room temperature, and alkalinized in iodoacetamide (3 mM) in the dark at room temperature for 15 min. The tryptic digestion protocol was performed using the KingFisher DuoPrime purification system (Thermo Fisher Scientific) in a series of steps. First, magnetic hydrophobic and hydrophilic beads were washed several times in MS-grade water and added to the deepwell plate in the KingFisher along with the samples and the same volume as the sample of 100% ethanol. Next, the solutions were mixed at low speed for 10 min, after which the beads coupled to the proteins were collected with the magnetic arm of the KingFisher and transferred to be washed in three different deepwells each containing 80% of ethanol. The washed beads–proteins were then released into the trypsin (V5111; Promega)-containing deepwells at a 50:1 (w/w) protein-to-protease ratio and mixed at low speed for 8 h of digestions into peptide fragments at 37°C in the KingFisher. Peptide samples were transferred into low protein binding tubes; 1% of TFA was added to acidify the samples ready to be desalted, cleaned, and concentrated on C18 tips (87784; Thermo Fisher Scientific) (Rappsilber et al, 2007) according to the manufacturer's instructions. Purified peptides were dried and resuspended in low protein binding tubes before MS analysis in 30 μl of 0.15% TFA and 1% acetic acid in MS-grade water.

### MS

The peptides were analyzed using a MS shotgun proteomics technique. This technique allows a sensitive bottom-up approach that consists of separating peptides resulting from protein digestion by liquid HPLC followed by tandem mass spectrometry (MS/MS). Samples were run on a Bruker timsTOF Pro mass spectrometer connected to an Evosep One liquid chromatography system. Tryptic peptides were resuspended in 0.1% formic acid, and each sample was loaded onto an Evosep tip. The Evosep tips were placed in position on the Evosep One, in a 96-tip box. The autosampler is configured to pick up each tip, elute, and separate the peptides using a set chromatography method (Bache et al, 2018). The chromatography buffers used were buffer B (99.9% acetonitrile, 0.1% formic acid) and buffer A (99.9% water, 0.1% formic acid). All solvents are LC-MS–grade.

The mass spectrometer was operated in a positive ion mode with a capillary voltage of 1,500 V, dry gas flow of 3 liters/min, and a dry temperature of 180°C. All data were acquired with the instrument operating in a trapped ion mobility spectrometry mode. Trapped ions were selected for ms/ms using parallel accumulation–serial fragmentation. A scan range of (100–1,700 m/z) was performed at a rate of 5 parallel accumulation–serial fragmentation MS/MS frames to 1 MS scan with a cycle time of 1.03 s (Meier et al, 2018).

The data analysis was done using MaxQuant software (Cox & Mann, 2008). The raw data were searched against the *Homo sapiens* subset of the UniProt/SwissProt database (reviewed) with the search engine MaxQuant (release 2.0.3.0). Specific parameters for trapped ion mobility spectrometry data–dependent acquisition were used: Fixed Mod: carbamidomethylation; Variable Mods: methionine, oxidation; Trypsin/P digest enzyme: maximum two missed cleavages; Precursor mass tolerances: 10 ppm; Peptide FDR: 1%; and Protein FDR: 1%. The normalized protein intensity of each identified protein was used for label-free quantitation (LFQ) using the MaxLFQ algorithm (Cox et al, 2014).

The MS proteomics data have been deposited to the ProteomeXchange Consortium via the PRIDE (Perez-Riverol et al, 2022) partner repository with the dataset identifier PXD035399.

## AP-MS data filtering and ID mapping

The data were first filtered based on the label-free quantification intensities (LFQi) using the following five steps: (i) removal of proteins that were labeled as "only identified by site", "potential contaminant", and "reverse"; (ii) removal of all observations with LFQi equals to 0; (iii) removal of outlier samples (based on low overall LFQi; see Fig S3); (iv) removal of proteins that are not present in at least 60% of the samples of a group for each group (a group is defined as the collection of three biological with two technical replicates for one condition, which results in a group size of maximum 6); and (v) filtering against the negative control sample, which is only the beads used for the AP-MS sample preparations, by only considering proteins for further analysis that are significantly higher found in the samples compared with the negative control. In MS analysis–based proteomics data, there are typically two types of missing values, the missing not at random (MNAR) and the missing at random (MAR) (Lazar et al, 2016). A mixed imputation strategy was chosen, with kNN imputation as the strategy for MAR values (Gatto & Lilley, 2012; Gatto et al, 2021; Rainer et al, 2022). Other missing values were considered MNAR values and imputed at value 0. After the imputation, differential interaction analysis was performed for each group against the bead control. $P$-values were adjusted using FDR correction as described by Benjamini and Hochberg (1995). Afterward, all proteins were extracted for each group, which were significantly enriched in the sample (cutoffs: $P$-value–adjusted: <0.01, log fold change: >1). The data were transformed to have consistent protein and gene name annotations after the data filtering. The data are received from MaxQuant software in UniProt IDs and mapped to HGNC gene names using the HGNC database (retrieved 12/2021). However, one UniProt ID can correspond to multiple HGNC gene names. In this case, manual selection of the gene names of interest was performed. Finally, the HGNC names were mapped to gene IDs of the SysGO database (Luthert & Kiel, 2020). A couple of proteins could not be found in the SysGO database, and one protein was renamed (i.e., HGNC name: PHB1, which was renamed PHD for SysGO). Then, the technical replicates were merged using the median. In summary, we obtain a dataset with raw LFQi (Table S2) or log$_2$-transformed (Table S3) data with biological triplicates. Data preparation was performed in R (http://www.r-project.org/index.html) using the following packages: dplyr (Beckerman et al, 2017), tidyr (Wickham et al, 2019), stringr (Wickham, 2010), tidyxl, purr (Mailund, 2019), DEP (Zhang et al, 2018), and limma (Ritchie et al, 2015; Phipson et al, 2016). The script file for the data preparation and the data pre- and post-preparation are available on Zenodo (Camille et al, 2022).

## Functional analysis of the interactome

Functional analysis of the interactome was performed in two different ways. The first approach consists of a differential interaction analysis based on the filtered LFQ intensities. Imputation was performed by a mixed imputation strategy, using bpca (Bayesian PCA) (Oba et al, 2003; Stacklies et al, 2007) for MNAR values and MinProb (https://cran.r-project.org/web/packages/imputeLCMD/index.html) for MAR values. Differential analysis was performed using limma (Ritchie et al, 2015; Phipson et al, 2016) and DEP (Zhang et al, 2018). $P$-values were adjusted using FDR correction by Benjamini and Hochberg (1995). The results of the differential interaction analysis were evaluated for functional enrichment by performing a gene set enrichment analysis using ClusterProfiler4 (Yu et al, 2012; Wu et al, 2021) against the GO biological process (Ashburner et al, 2000; Gene Ontology Consortium, 2021).

For the second approach, LFQ intensities were collapsed on GO BP terms by summing up all intensities of all identified proteins for each sample for each GO term. Then, for each GO term, a three-way ANOVA was performed with the main effects of mutation status (genetic context), condition and concentration (culture context), and their interaction terms. The $P$-values of these ANOVAs were collectively corrected using correction by Holm (1979). After correction, significant terms ($P < 0.05$) were further analyzed using Tukey's Honest Significant Difference post hoc tests. $P$-values were collectively corrected using FDR correction by Benjamini and Hochberg (1995).

Both approaches identify many GO terms that are significantly different ($P < 0.05$ after respective adjustment) between the groups. To gain an overview over the results, the semantic similarity between the GO terms was determined using the methodology proposed by Schlicker et al (2006) and Yu et al (2010). Based on the resulting similarity matrix, GO terms were clustered using the binary cut algorithm (Gu & Hübschmann, 2022b). The results were visualized as a heatmap with data from the analysis projected as additional heatmaps (Gu et al, 2016; Gu & Hübschmann, 2022b).

All analysis in this part was performed using the R programming language and the tidyverse environment (Wickham et al, 2019). The scripts and output for this analysis are available on Zenodo (Camille et al, 2022).

## Visualization of AP-MS data in the Shiny app

The results from the functional analysis together with the filtered AP-MS data were put together in an R Shiny dashboard, allowing the interactive exploration of our analysis and data (Sievert, 2020; Gu & Hübschmann, 2022a). The R Shiny app is available at https://pjunk.shinyapps.io/kras_apms_vis/, with the source code and underlying data files available at https://github.com/PhilippJunk/kras_apms_vis

## Assessment of phenotypic and metabolic parameters of Caco-2 cells

Caco-2 cells cultured in DMEM supplemented with 10% FBS were seeded at about 70% confluency in nine 12-well plates (CELLSTAR, Greiner Bio-One) to test three KRAS mutant status (WT, G12D, and G12C) in three contexts (unstimulated, DMOG, and IL-6). 24-h post-seeding cells were transfected with FLAG-KRASWT, FLAG-KRASG12D, or FLAG-KRASG12C with the protocol previously described. Then, 5-h post-transfection medium was changed and replaced with DMEM supplemented with 1% glutamine containing either 20 ng/ml DMOG, 20 ng/ml IL-6, or no stimulus (unstimulated). In addition, for cells transfected with KRASWT, 15 ng/ml doxycycline was added for

plasmid activation. Cell suspension samples were collected: once for all groups during the seeding (24 h before transfection), then in triplicate for each group at 24, 48, and 72 h post-transfection. Medium samples were collected: once during the context introduction (5 h post-transfection), then in triplicate for each group at 24, 48, and 72 h post-transfection. Cell suspension samples were used for cell counting using Scepter 2.0 Automated Cell Counter with 60-$\mu$m sensors (Merck Millipore), for cell viability and cellular ATP assessments using CellTiter-Glo Luminescent Cell Viability Assay (Promega), and for Western blots of FLAG-KRAS (Anti-FLAG M2, F3165, mouse/monoclonal, 1:1,000 dilution; Sigma-Aldrich) normalized with $\beta$-actin (#4970, rabbit/monoclonal, 1:3,000 dilution; Cell Signaling) using the protocol previously described. For the Western blot, after cell lysis, for each plasmid and at each time, the three context replicates were pooled to obtain enough protein to prepare 40 $\mu$l loading solution at 0.25 $\mu$g/$\mu$l (e.g., for WT at 24 h, the three replicates are one Unstim, one DMOG, and one IL-6 sample). Media were used for the assessment of glucose uptake and lactate release using, respectively, Glucose-Glo and Lactate-Glo assays (Promega).

### Statistical analysis

All data are expressed as the average ± SD, with SD represented by error bars. Statistical comparisons between two groups (typically treated group against control samples) were performed using a $t$ test. The average value and SD were calculated from at least three biological experiments. All tests were performed with a $P$-value of 0.05 using GraphPad Prism 9 software.

### Network reconstruction and random walk analysis of AP-MS data

The starting point of the network is the 56 potential effectors of KRAS (Ibáñez Gaspar et al, 2021). Then, beginning from these effectors, STRING (version 11.5) was used to construct the network (Szklarczyk et al, 2019). All nodes that had a shortest path of 4 or less to these effectors were included, while filtering out edges with a STRING confidence score of less than 0.7. For KRAS, only edges toward the effectors were included in the network. Apart from the KRAS–effector edges, all interactions in the network are considered undirected. The final network consists of 15,062 nodes and 493,838 edges.

Using the network, targeted random walks were performed starting from KRAS, in the following called source, for each target protein in each condition of interest. For a predetermined number of steps, based on the current node, one of the connected nodes is randomly chosen. For each random walk, there is always only one target protein. As soon as the target protein is reached, or a certain number of iterations have been exceeded, the walk ends. The random walks are biased toward proteins found in the interactome of a certain condition. This is facilitated by favoring nodes found in the interactome by a factor of 20 over nodes not found in the interactome of the specific condition. The actual probability depends on the number of connecting nodes.

$$P(in\ APMS) = 20 * P(not\ in\ APMS) .$$

The number of iterations for the random walk for each target is dynamically calculated based on the length of the shortest path between the source and the target.

$$walklen = shortestpath + 2 .$$

Finally, the number of random walks, limited by runtime and memory, was set to 100,000,000 for each target for each condition. The code for the random walks was written in python using NumPy, SciPy, pandas, Numba, and CrsGraph. The script used to run this analysis is available on Zenodo (Camille et al, 2022).

Analysis of the random walks was performed by filtering out any paths that were found less than 100/100,000,000 walks and selecting the top 10 identified paths for each target by frequency. Paths were decomposed into a sequence of edges, and all edges for one condition were concatenated to generate a condition-specific network of information flow from KRAS to all proteins associated with a specific GO term. Networks were visualized, and the effector layer of each network was extracted and visualized together.

All analysis and visualization were performed using R, in particular, the packages dplyr, tidyr, stringr, purrr, furrr, ggplot, and ggraph (Wickham, 2010; Wilkinson, 2011; Mailund, 2019; Wickham et al, 2019; Pedersen, 2020).

### Proximity ligation assay

PLA was performed to evaluate the recruitment of p110α-PI3K to Ras, using NaveniFlex MR (Navinci Diagnostics). First, Caco-2 cells were seeded at a confluency of 35,000 cells/cm2 in eight-well chamber slides, and left to adhere overnight. Cells were then serum-starved in DMEM+GlutaMAX supplemented with 0.2% FBS for 24 h, at 37°C and 5% $CO_2$. Cells were then stimulated with 20 ng/ml EGF for either 0, 5, 15, or 45 min, after which media were removed and cells were washed with ice-cold PBS. Then, cells were fixed in 3.7% formaldehyde in PBS solution for 15 min on ice. Slides were then washed for 3 × 5 min in PBS, dried, and permeabilized in a 0.2% Triton X-100/TBS solution for 10 min at room temperature. Proximity ligation assay was then performed as described in Wåhlén et al (2022). Primary antibodies used were as follows: rabbit anti-Ras (52939; Abcam) at 1:200, and mouse anti-p110α-PI3K (611399; BD Biosciences) at 1:50. Image analysis was performed as described in Wåhlén et al (2022) with the exception that pictures were taken in replicates of five per experimental condition, and a 63×/1.4 oil objective was used. Images have been enhanced for visualization purposes, but the image analysis has been performed on original images.

## Data Availability

All data are available in the main text or the supplementary materials. The MS proteomics data have been deposited to the ProteomeXchange Consortium via the PRIDE (Perez-Riverol et al, 2022) partner repository with the dataset identifiers PXD035399 and PXD039404. Data processing and analysis pipeline and results are available at Zenodo

(Camille et al, 2023). Access to the AP-MS data through the Shiny app is available via GitHub ([https://github.com/PhilippJunk/kras_apms_vis](https://github.com/PhilippJunk/kras_apms_vis)).

# Supplementary Information

# Acknowledgements

The authors would like to thank all members of the Kiel laboratory for discussions and critical reading of the article. This work received funding from the Science Foundation Ireland Grant 16/FRL/3886 (to C Kiel) and from the Comprehensive Molecular Analytical Platform (CMAP) under the SFI Research Infrastructure Programme 18/RI/5702 (to K Wynne).

## Author Contributions

C Ternet: formal analysis, investigation, methodology, and writing—original draft.
P Junk: software, formal analysis, investigation, methodology, and writing—review and editing.
T Sevrin: investigation and writing—review and editing.
S Catozzi: data curation and writing—review and editing.
E Wåhlén: investigation and methodology.
J Heldin: investigation and methodology.
G Oliviero: investigation, methodology, and writing—review and editing.
K Wynne: resources, investigation, methodology, and writing—review and editing.
C Kiel: conceptualization, data curation, supervision, funding acquisition, methodology, and writing—original draft.

## Conflict of Interest Statement

The authors declare that they have no conflict of interest.

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
