## [Reviewer comments · Life Science Alliance]

Analysis of context-specific KRAS-effectors (sub)complexes in Caco-2 cells

Camille Ternet, Philipp Junk, Thomas Sevrin, Simona Catozzi, Erik Wåhlén, Johan Heldin, Giorgio Oliviero, Kieran Wynne, and Christina Kiel

DOI: <https://doi.org/10.26508/lsa.202201670>

Corresponding author(s): Christina Kiel, University of Pavia

Review Timeline:

Submission Date:	2022-08-12
Editorial Decision:	2022-09-12
Revision Received:	2023-01-13
Editorial Decision:	2023-02-08
Revision Received:	2023-02-22
Editorial Decision:	2023-02-22
Revision Received:	2023-02-24
Accepted:	2023-02-27

Transaction Report:

September 12, 2022

Re: Life Science Alliance manuscript #LSA-2022-01670-T

Prof Christina Kiel
University of Pavia
Molecular Medicine
Via Forlanini 6
Pavia, Lombardia 27100
Italy

Dear Dr. Kiel,

Thank you for submitting your manuscript entitled "Analysis of context-specific KRAS-effectors (sub)complexes in Caco-2 cells" to Life Science Alliance. The manuscript was assessed by expert reviewers, whose comments are appended to this letter. We invite you to submit a revised manuscript addressing the Reviewer comments.

Thank you for this interesting contribution to Life Science Alliance. We are looking forward to receiving your revised manuscript.

Sincerely,

B. MANUSCRIPT ORGANIZATION AND FORMATTING:

Reviewer #1 (Comments to the Authors (Required)):

This paper investigates exogenous KRAS variant protein-protein interactions in the presence of the wild type allele in caco-2 cells under varied culture conditions by FLAG affinity purification mass spectrometry using a label-free quantification method and initiates analysis of the functional consequences of various sub-complexes. This work specifically advances our understanding of KRAS interactions under physiologically relevant culture conditions with various levels of growth factor stimulation or hypoxia. The novelty of this work is the method used to investigate KRAS interactions, the culture conditions used to stimulate complex formation, and the bioinformatic analysis of the interaction networks. Overall, the major claims are substantiated, and data are clearly presented. However, the wording of several claims should be modified/clarified, and a few experiments should be added to strengthen the paper. The proposed experiments could be run in several weeks. Please see comments below.

Claim 1: "The LFQ intensity of the KRAS bait is comparable across all AP-MS conditions (Fig. S4), which suggests that a similar quantity of FLAG-KRAS proteins binds to the magnetic beads across all APs."

This data supports a recurrent theme throughout the paper. When you say LFQ intensity of the KRAS bait, are you referring to FLAG peptide abundance in the enrichments or is this the combined expression of KRAS peptides (both endogenous and exogenous)? See claim 3 and claim 5 below.

Claim 2: "Altogether, we provide a near-to-complete binding landscape of effectors in complex with KRAS under the conditions tested."

It is interesting to note that you identified significantly more KRAS interactions in comparison to previous reports, could you clarify why you used a 60% threshold across technical and biological replicates to identify high confidence interactions?

Claim 3: "Taken together, these results suggest that the proteins detected in complex with KRAS seem to be more conditions-dependent rather than mutation-dependent."

This is supported by the experimental data and uMAP/PCA analysis. However, it would be greatly appreciated if the authors could provide western blots for Flag KRAS variants under each experimental condition used (see claim 5, below). This would significantly strengthen your paper. As noted in the discussion, it would also be helpful to measure the relative abundance of some of the condition specific interactions that you identified, as opposed to basal mRNA abundance in Caco-2 taken from the human protein atlas. This would help establish whether the differences in complex composition stem from

Claim 4: "All together, this supports our initial hypothesis that group 2 effectors tend to be found in complex with Ras only in specific conditions that promote PM recruitment via RBD- independent mechanisms."

This is supported by computational analysis. Experimental validation showing increased PM colocalization via microscopy in Caco-2 cells would better support this claim.

Claim 5: "Those results suggest that for all genetic contexts, exogenous KRAS was effectively expressed during a 72 h period. However, we also noticed that exogenous KRAS expression was significantly different between the three genetic contexts."

Why is there such a big difference in expression across variants? In the figure legend for S10 you say that these samples were from "a pool of the 3 context samples (1 Unstim, 1 DMOG and 1 IL-6 sample)." It would be greatly appreciated to show the expression differences in each condition as a separate western blot (As noted above in claim 3 and 5).

Claim 6: "It also suggests that changes on the level of the interactome, particularly those seen with different culture (microenvironmental) contexts, likely manifest as phenotypic change"

This claim overreaches. The culture conditions could change metabolic phenotype through numerous different mechanisms. To make the claim that KRAS interactions manifest as phenotypic changes, you could inhibit KRAS signaling in each experimental condition or disrupt individual complex components using genetic methods. Based on the data, you could state that the bioinformatic analysis of KRAS complex composition correlates with the phenotypic changes.

Claim 7: "This implies that the results of the functional analysis based on AP-MS data are more suitable at capturing changes in metabolism at the level of the whole cell culture than at the level of a single cell."

This is an interesting finding. Do you have a hypothesis?

Reviewer #2 (Comments to the Authors (Required)):

The manuscript by Ternet et al. seeks to determine RAS mutation specific differences in different conditions and thus explored RAS interactome components using affinity purification based mass-spectrometry approach. The authors then used bioinformatics approach and determined a set of potential RAS-effector protein interactions that differentially associate in different conditions but are not specific to RAS mutations. There is potentially an interesting story here, but it is currently under-developed. Without significant additional experimental validation on the primary proteomics findings of the potential effector-KRAS interactions in different contexts, the impact of this effort on the field is may be rather limited.

Specific comments:

1. Without any experimental evaluation of any of the identified changes in the interactome are perhaps less than interpretable and lose value as a general data resource.
2. The conditions used to evaluate the RAS interactome is rather an artificial setting and may not truly represent a physiologically relevant setting.
3. The differences in KRAS protein levels (Fig S10) in the different conditions could simply reflect the affinity capture of KRAS protein at differential efficiency and as a consequence results in changes in interactome components rather than signaling changes.
4. Similarly, it is possible that differences in expression of interactome components across treatment groups reflect the affinity capture efficiency. Targeted experimental validation of mRNA/protein expression levels of the identified potential hits should be utilized as a proof-of-principle test. If this is the case then it does raise questions regarding the value of the proteomic datasets relevant to different conditions.

Reviewer #1 (Comments to the Authors (Required)):

This paper investigates exogenous KRAS variant protein-protein interactions in the presence of the wild type allele in caco-2 cells under varied culture conditions by FLAG affinity purification mass spectrometry using a label-free quantification method and initiates analysis of the functional consequences of various sub-complexes. This work specifically advances our understanding of KRAS interactions under physiologically relevant culture conditions with various levels of growth factor stimulation or hypoxia. The novelty of this work is the method used to investigate KRAS interactions, the culture conditions used to stimulate complex formation, and the bioinformatic analysis of the interaction networks. Overall, the major claims are substantiated, and data are clearly presented. However, the wording of several claims should be modified/clarified, and a few experiments should be added to strengthen the paper. The proposed experiments could be run in several weeks. Please see comments below.

Comment 1: *"Claim 1: "The LFQ intensity of the KRAS bait is comparable across all AP-MS conditions (Fig. S4), which suggests that a similar quantity of FLAG-KRAS proteins binds to the magnetic beads across all APs." This data supports a recurrent theme throughout the paper. When you say LFQ intensity of the KRAS bait, are you referring to FLAG peptide abundance in the enrichments or is this the combined expression of KRAS peptides (both endogenous and exogenous)? See claim 3 and claim 5 below."*

Response 1: It is the expression of the exogenous (Flag-) KRAS peptides. We mention this now in the legend of Fig. S6.

Comment 2: *"Claim 2: "Altogether, we provide a near-to-complete binding landscape of effectors in complex with KRAS under the conditions tested." It is interesting to note that you identified significantly more KRAS interactions in comparison to previous reports, could you clarify why you used a 60% threshold across technical and biological replicates to identify high confidence interactions?"*

Response 2: The 60 % threshold was chosen in order to balance false-positive (detection of proteins that are not interactors) and false-negative (missing at random values) interactions in our data set. We believe that observing a protein in at least 4 out of 6 (technical and biological) replicates should give us good confidence in the presence of this interaction in the respective condition.

Comment 3: *"Claim 3: "Taken together, these results suggest that the proteins detected in complex with KRAS seem to be more conditions-dependent rather than mutation-dependent." This is supported by the experimental data and uMAP/PCA analysis. However, it would be greatly appreciated if the authors could provide western blots for Flag KRAS variants under each experimental condition used (see claim 5, below). This would significantly strengthen your paper. As noted in the discussion, it would also be helpful to measure the relative abundance of some of the condition specific*

interactions that you identified, as opposed to basal mRNA abundance in Caco-2 taken from the human protein atlas. This would help establish whether the differences in complex composition stem from"

Response 3: We provide these western blots now for Flag-KRAS G12D grown in all 'culture contexts' (Fig. S3). We have also performed mass spectrometry analyses to assay the abundances of effectors in different 'culture contexts' before AP-MS (Fig. S4 and S8).

Comment 4: "Claim 4: "All together, this supports our initial hypothesis that group 2 effectors tend to be found in complex with Ras only in specific conditions that promote PM recruitment via RBD- independent mechanisms."
This is supported by computational analysis. Experimental validation showing increased PM colocalization via microscopy in Caco-2 cells would better support this claim."

Response 4: In the experimental AP-MS setup used in this work, exogenous expressions of Flag-KRAS WT and mutant variants are used. Techniques that assay protein-protein interactions based on co-localization use endogenous proteins that are detected using antibodies, for example proximity ligation assays (PLA). While AP-MS experiments can be performed in a (semi)-high-throughput fashion, PLA assays cannot as high quality antibodies are needed and substantial experimental optimization is needed on a case-by-case (for each effector) basis. Furthermore, PLA is typically not performed on exogenously expressed proteins as it is difficult to account for cell to cell expression variation. Nevertheless, we have performed PLA assays on WT Caco-2 cells and the effector PI3K with the stimulus EGF, where we show increased Ras/PM co-localization, which however changes over time. We use these results in the discussion to highlight pros and cons of different methods for assaying protein-protein interactions.

Comment 5: "Claim 5: "Those results suggest that for all genetic contexts, exogenous KRAS was effectively expressed during a 72 h period. However, we also noticed that exogenous KRAS expression was significantly different between the three genetic contexts."
Why is there such a big difference in expression across variants? In the figure legend for S10 you say that these samples were from "a pool of the 3 context samples (1 Unstim, 1 DMOG and 1 IL-6 sample)." It would be greatly appreciated to show the expression differences in each condition as a separate western blot (As noted above in claim 3 and 5)."

Response 5: We provide these western blots now for Flag-KRAS G12D grown in all 'culture contexts' (Fig. S3). The expression levels across conditions are very similar. As to the why there are different expression levels between the mutants, this could be related to differences in translation (e.g. tRNA levels) or to differences on the level of protein degradation. Indeed, we have shown earlier that different doxycycline levels are needed in order to reach similar protein levels of WT and G12D (Beltran-Sastre et al., 2015; PMID: 26612112).

Comment 6: *"Claim 6: "It also suggests that changes on the level of the interactome, particularly those seen with different culture (microenvironmental) contexts, likely manifest as phenotypical change"*

This claim overreaches. The culture conditions could change metabolic phenotype through numerous different mechanisms. To make the claim that KRAS interactions manifest as phenotypic changes, you could inhibit KRAS signaling in each experimental condition or disrupt individual complex components using genetic methods. Based on the data, you could state that the bioinformatic analysis of KRAS complex composition correlates with the phenotypic changes."

Response 6: We have reframed this in the way suggested.

Comment 7: *"Claim 7: "This implies that the results of the functional analysis based on AP-MS data are more suitable at capturing changes in metabolism at the level of the whole cell culture than at the level of a single cell."*

This is an interesting finding. Do you have a hypothesis?"

Response 7: Our hypothesis is that it may be related to metabolic dependencies between cells, where lactate often plays a role. However, we prefer to not add additional hypothesis at this point of the results chapter.

Reviewer #2 (Comments to the Authors (Required)):

The manuscript by Ternet et al. seeks to determine RAS mutation specific differences in different conditions and thus explored RAS interactome components using affinity purification based mass-spectrometry approach. The authors then used bioinformatics approach and determined a set of potential RAS-effector protein interactions that differentially associate in different conditions but are not specific to RAS mutations. There is potentially an interesting story here, but it is currently under-developed. Without significant additional experimental validation on the primary proteomics findings of the potential effector-KRAS interactions in different contexts, the impact of this effort on the field is may be rather limited.

Specific comments:

Comment 8: *"1. Without any experimental evaluation of any of the identified changes in the interactome are perhaps less than interpretable and lose value as a general data resource."*

Response 8: We have already performed phenotype analyses for some of the conditions in the results chapter "Analysis of cell phenotypes in selected genetic and culture contexts"

Comment 9: *"2. The conditions used to evaluate the RAS interactome is rather an artificial setting and may not truly represent a physiologically relevant setting."*

Response 9: We have reframed this sentence in the introduction into "Our study provides an in-depth reconstruction of PPI networks mediated by oncogenic KRAS-

effector proteins in culture contexts that mimic some aspects of (patho)physiological colon contexts”.

Comment 10: “3. *The differences in KRAS protein levels (Fig S10) in the different conditions could simply reflect the affinity capture of KRAS protein at differential efficiency and as a consequence results in changes in interactome components rather than signaling changes.*”

Response 10: Fig S10 (now Fig. S13) actually shows a time course of Flag-KRAS expression over the time where phenotypic assays are conducted. The amount of Flag-KRAS protein captured in the AP-MS experiment is actually shown in Fig. S6 (previous Fig. S4), which is relatively consistent across all culture and genetic contexts.

Comment 11: “4. Similarly, it is possible that differences in expression of interactome components across treatment groups reflect the affinity capture efficiency. Targeted experimental validation of mRNA/protein expression levels of the identified potential hits should be utilized as a proof-of-principle test. If this is the case then it does raise questions regarding the value of the proteomic datasets relevant to different conditions.”

Response 11: We now provide additional data by western blotting showing for one genetic context (Flag-KRAS G12D) grown in all ‘culture contexts’ that exogenous expression levels are not affected by the culture condition (Fig. S3). We have also performed additional mass spectrometry analyses to assay the abundances of effectors in different ‘culture contexts’ before AP-MS (Fig. S4 and S8), which are in the same order of magnitude expressed across the different culture conditions.

February 8, 2023

Re: Life Science Alliance manuscript #LSA-2022-01670-TR

Prof. Christina Kiel
University of Pavia
Molecular Medicine
Via Forlanini 6
Pavia, Lombardia 27100
Italy

Dear Dr. Kiel,

Thank you for submitting your revised manuscript entitled "Analysis of context-specific KRAS-effectors (sub)complexes in Caco-2 cells" to Life Science Alliance. The manuscript has been seen by the original reviewers whose comments are appended below. While the reviewers continue to be overall positive about the work in terms of its suitability for Life Science Alliance, some important issues remain.

Our general policy is that papers are considered through only one revision cycle; however, given that the suggested changes are relatively minor, we are open to one additional short round of revision. Please note that I will expect to make a final decision without additional reviewer input upon re-submission.

Please submit the final revision within one month, along with a letter that includes a point by point response to the remaining reviewer comments.

To upload the revised version of your manuscript, please log in to your account: <https://lsa.msubmit.net/cgi-bin/main.plex>
You will be guided to complete the submission of your revised manuscript and to fill in all necessary information.

B. MANUSCRIPT ORGANIZATION AND FORMATTING:

Sincerely,

Reviewer #1 (Comments to the Authors (Required)):

After review of the updated manuscript, the expression of exogenous KRAS under the various culture contexts remains obscure. As this experiment impacts the bioinformatic analysis and interpretation of the data set, this should be clarified (experimentally or computationally) prior to publication, in addition to modifying the claims based on this experiment, correcting errors in figure legends and grammatical issues outlined below.

Comment 1: "Claim 1: "The LFQ intensity of the KRAS bait is comparable across all AP-MS conditions (Fig. S4), which suggests that a similar quantity of FLAG-KRAS proteins binds to the magnetic beads across all APs."

This data supports a recurrent theme throughout the paper. When you say LFQ intensity of the KRAS bait, are you referring to FLAG peptide abundance in the enrichments or is this the combined expression of KRAS peptides (both endogenous and exogenous)? See claim 3 and claim 5 below."

Response 1: It is the expression of the exogenous (Flag-) KRAS peptides. We mention this now in the legend of Fig. S6.

Reviewer 1 response to comment 1: Ok, thank you for clarifying. Please see comment 3.

Comment 2: "Claim 2: "Altogether, we provide a near-to-complete binding landscape of effectors in complex with KRAS under the conditions tested."

It is interesting to note that you identified significantly more KRAS interactions in comparison to previous reports, could you clarify why you used a 60% threshold across technical and biological replicates to identify high confidence interactions?"

Response 2: The 60 % threshold was chosen in order to balance false-positive (detection of proteins that are not interactors) and false-negative (missing at random values) interactions in our data set. We believe that observing a protein in at least 4 out of 6 (technical and biological) replicates should give us good confidence in the presence of this interaction in the respective condition.

Reviewer 1 response to comment 2: Ok, this is now explicitly stated in the results section. In your methods section, could you also clarify the threshold value used to filter against your negative control (beads only)?

Comment 3: "Claim 3: "Taken together, these results suggest that the proteins detected in complex with KRAS seem to be more conditions-dependent rather than mutation-dependent."

This is supported by the experimental data and uMAP/PCA analysis. However, it would be greatly appreciated if the authors could provide western blots for Flag KRAS variants under each experimental condition used (see claim 5, below). This would significantly strengthen your paper. As noted in the discussion, it would also be helpful to measure the relative abundance of some of the condition specific interactions that you identified, as opposed to basal mRNA abundance in Caco-2 taken from the human protein atlas. This would help establish whether the differences in complex composition stem from"

Response 3: We provide these western blots now for Flag-KRAS G12D grown in all 'culture contexts' (Fig. S3). We have also performed mass spectrometry analyses to assay the abundances of effectors in different 'culture contexts' before AP-MS (Fig. S4 and S8).

Reviewer 1 response to comment 3: Based on the pan-RAS staining (Fig S3) and mass spectrometry (Fig S4), differential expression of exogenous RAS or the effectors does not correlate with context selective effector enrichment. However, you do not show anti-Flag staining in Fig S3A, as you did in Fig. S2A and S13, which would demonstrate how much exogenous KRAS is expressed in the various culture conditions prior to enrichment. Although you show the bait intensity after enrichment in Fig S6, based on this quantification (Fig S6), the KRAS bait abundance significantly correlates with the number of proteins enriched in each culture condition (Fig 1A) and the selective enrichment of effectors (Fig 3A/B). This relationship can be validated by normalizing LFQ intensity to the KRAS bait using dataset S1 or S2, and matches the binding affinity of effector proteins previously quantified in the literature. Alternatively, plot the KRAS (Fig S3E, and Fig S6) and effector protein expression values (Fig S4) consistently for comparison (either Log₂(LFQ) or raw LFQ), or show the exogenous KRAS expression from the unenriched proteomic dataset (Fig S4). Moreover, the figure legend in Fig S6 states that the plot shows "Log 10-fold change LFQ", but the x axis of the plot is labeled "LFQ intensity". Additionally, the pan Ras staining in Fig S3 shows two bands in gel 1 and three bands in gel 2, whereas in Fig S2 and Fig S3 there is one primary band when exogenous KRAS is expressed, which is confusing. Although the legend of Fig S3 indicates that the additional bands are HRAS and NRAS, why are these additional bands not shown in the other gels? I realize that transient transfections are inherently variable, and you mention this in the discussion, but I think acknowledging that the exogenous FLAG-KRAS expression is dependent upon culture conditions, correlated with the number and content of enriched KRAS effectors, and that the phenotypic effects could be associated with abundance rather than complex composition, is warranted. Thus, modifying claims throughout the paper to acknowledge this relationship would be necessary. Alternatively, if exogenous KRAS expression under the various culture conditions prior to enrichment via anti-FLAG staining of the western blot (Fig S3A) or unenriched mass spectrometry (Fig S4) does not show any correlation with the effector enrichment or content, disregard these comments. Either way the data are informative, but this experiment significantly impacts interpretation of the results and the claims of your paper.

Comment 4: "Claim 4: "All together, this supports our initial hypothesis that group 2 effectors tend to be found in complex with

Ras only in specific conditions that promote PM recruitment via RBD- independent mechanisms."

This is supported by computational analysis. Experimental validation showing increased PM colocalization via microscopy in Caco-2 cells would better support this claim."

Response 4: In the experimental AP-MS setup used in this work, exogenous expressions of Flag-KRAS WT and mutant variants are used. Techniques that assay protein-protein interactions based on co-localization use endogenous proteins that are detected using antibodies, for example proximity ligation assays (PLA). While AP-MS experiments can be performed in a (semi)-high-throughput fashion, PLA assays cannot as high quality antibodies are needed and substantial experimental optimization is needed on a case-by-case (for each effector) basis. Furthermore, PLA is typically not performed on exogenously expressed proteins as it is difficult to account for cell to cell expression variation. Nevertheless, we have performed PLA assays on WT Caco-2 cells and the effector PI3K with the stimulus EGF, where we show increased Ras/PM co-localization, which however changes over time. We use these results in the discussion to highlight pros and cons of different methods for assaying protein-protein interactions.

Reviewer 1 response to comment 4: The PLA analysis seems to support your hypothesis, but relevant controls are lacking. For example, a negative control treatment in addition to the EGF stimulation over the time course. You imply that this control was included in the figure legend for S18, but do not show this data. Without this control, you cannot tell whether the signal is background variation over the timecourse.

Comment 5: "Claim 5: "Those results suggest that for all genetic contexts, exogenous KRAS was effectively expressed during a 72 h period. However, we also noticed that exogenous KRAS expression was significantly different between the three genetic contexts."

Why is there such a big difference in expression across variants? In the figure legend for S10 you say that these samples were from "a pool of the 3 context samples (1 Unstim, 1 DMOG and 1 IL-6 sample)." It would be greatly appreciated to show the expression differences in each condition as a separate western blot (As noted above in claim 3 and 5)."

Response 5: We provide these western blots now for Flag-KRAS G12D grown in all 'culture contexts' (Fig. S3). The expression levels across conditions are very similar. As to the why there are different expression levels between the mutants, this could be related to differences in translation (e.g. tRNA levels) or to differences on the level of protein degradation. Indeed, we have shown earlier that different doxycycline levels are needed in order to reach similar protein levels of WT and G12D (Beltran-Sastre et al., 2015; PMID: 26612112).

Reviewer 1 response to comment 5: Ok, thanks.

Comment 6: "Claim 6: "It also suggests that changes on the level of the interactome, particularly those seen with different culture (microenvironmental) contexts, likely manifest as phenotypical change"

This claim overreaches. The culture conditions could change metabolic phenotype through numerous different mechanisms. To make the claim that KRAS interactions manifest as phenotypic changes, you could inhibit KRAS signaling in each experimental condition or disrupt individual complex components using genetic methods. Based on the data, you could state that the bioinformatic analysis of KRAS complex composition correlates with the phenotypic changes."

Response 6: We have reframed this in the way suggested.

Reviewer 1 Response: Ok, thanks.

Comment 7: "Claim 7: "This implies that the results of the functional analysis based on AP-MS data are more suitable at capturing changes in metabolism at the level of the whole cell culture than at the level of a single cell."

This is an interesting finding. Do you have a hypothesis?"

Response 7: Our hypothesis is that it may be related to metabolic dependencies between cells, where lactate often plays a role. However, we prefer to not add additional hypothesis at this point of the results chapter.

Reviewer 1 Response: Ok, thanks.

General Comments:

There are many grammatical errors that need to be fixed in addition to the claims about culture context dependent KRAS complex composition. Specifically, in the results section and the discussion. Although this may be a language difference, in general, "mass Spectroscopy" is no longer used to describe "mass spectrometry" experiments, and should be changed throughout the paper.

Reviewer #2 (Comments to the Authors (Required)):

All my concerns have been addressed and the manuscript can be considered for publication.

Reviewer #1

Comment 1: *"After review of the updated manuscript, the expression of exogenous KRAS under the various culture contexts remains obscure. As this experiment impacts the bioinformatic analysis and interpretation of the data set, this should be clarified (experimentally or computationally) prior to publication, in addition to modifying the claims based on this experiment, correcting errors in figure legends and grammatical issues outlined below."*

Reply to comment 1: We appreciate the remaining comments raised by the reviewer. We have performed the analysis suggested by the reviewer and modified Fig. S6 (see reply to comment 4).

Original comment/response

Comment 1: "Claim 1: "The LFQ intensity of the KRAS bait is comparable across all AP-MS conditions (Fig. S4), which suggests that a similar quantity of FLAG-KRAS proteins binds to the magnetic beads across all APs."

This data supports a recurrent theme throughout the paper. When you say LFQ intensity of the KRAS bait, are you referring to FLAG peptide abundance in the enrichments or is this the combined expression of KRAS peptides (both endogenous and exogenous)? See claim 3 and claim 5 below."

Response 1: It is the expression of the exogenous (Flag-) KRAS peptides. We mention this now in the legend of Fig. S6.

Comment 2: *"Reviewer 1 response to comment 1: Ok, thank you for clarifying. Please see comment 3."*

Reply to comment 2: OK, thank you. We provide a reply to this comment below (reply to comment 4).

Original comment/response

Comment 2: "Claim 2: "Altogether, we provide a near-to-complete binding landscape of effectors in complex with KRAS under the conditions tested."

It is interesting to note that you identified significantly more KRAS interactions in comparison to previous reports, could you clarify why you used a 60% threshold across technical and biological replicates to identify high confidence interactions?"

Response 2: The 60 % threshold was chosen in order to balance false-positive (detection of proteins that are not interactors) and false-negative (missing at random values) interactions in our data set. We believe that observing a protein in at least 4 out of 6 (technical and biological) replicates should give us good confidence in the presence of this interaction in the respective condition.

Comment 3: "Reviewer 1 response to comment 2: Ok, this is now explicitly stated in the results section. In your methods section, could you also clarify the threshold value used to filter against your negative control (beads only)?"

Reply to comment 3: The information about the cut off used for filtering the beads control was already provided in the method section "AP-MS data filtering and ID mapping:..."(v) filtering against the negative control sample, which is only the beads used for the AP-MS sample preparations, by only considering proteins for further analysis that are significantly higher found in the samples compared to the negative control. In MS analysis based proteomic data, there are typically two types of missing values, the missing not at random (MNAR) and the missing at random (MAR) (Lazar et al, 2016). A mixed imputation strategy was chosen, with kNN imputation as the strategy for MAR values (Gatto & Lilley, 2012; Gatto et al, 2021; Rainer et al, 2022). Other missing values were considered MNAR values and imputed at value 0. After the imputation, differential interaction analysis was performed for each group against the beads control. P values were adjusted using FDR correction as described by (Benjamini & Hochberg, 1995). Afterward, all proteins were extracted for each group which were significantly enriched in the sample (cutoffs: p-value adjusted: < 0.01, Log Fold Change: > 1).".

Original comment/response

Comment 3: "Claim 3: "Taken together, these results suggest that the proteins detected in complex with KRAS seem to be more conditions-dependent rather than mutation-dependent."

This is supported by the experimental data and uMAP/PCA analysis. However, it would be greatly appreciated if the authors could provide western blots for Flag KRAS variants under each experimental condition used (see claim 5, below). This would significantly strengthen your paper. As noted in the discussion, it would also be helpful to measure the relative abundance of some of the condition specific interactions that you identified, as opposed to basal mRNA abundance in Caco-2 taken from the human protein atlas. This would help establish whether the differences in complex composition stem from"

Response 3: We provide these western blots now for Flag-KRAS G12D grown in all 'culture contexts' (Fig. S3). We have also performed mass spectrometry analyses to assay the abundances of effectors in different 'culture contexts' before AP-MS (Fig. S4 and S8).

Comment 4: "Reviewer 1 response to comment 3: Based on the pan-RAS staining (Fig S3) and mass spectrometry (Fig S4), differential expression of exogenous RAS or the effectors does not correlate with context selective effector enrichment. However, you do not show anti-Flag staining in Fig S3A, as you did in Fig. S2A and S13, which would demonstrate how much exogenous KRAS is expressed in the various culture conditions prior to enrichment. Although you show the bait intensity after enrichment in Fig S6, based on this quantification (Fig S6), the KRAS bait abundance significantly correlates with the number of proteins enriched in each culture condition (Fig 1A) and the selective enrichment of effectors (Fig 3A/B). This relationship can be validated by normalizing LFQ intensity to the KRAS bait using dataset S1 or S2, and matches the binding affinity of effector proteins previously quantified in the literature. Alternatively, plot the KRAS (Fig S3E, and Fig S6) and effector

protein expression values (Fig S4) consistently for comparison (either Log₂(LFQ) or raw LFQ), or show the exogenous KRAS expression from the unenriched proteomic dataset (Fig S4). Moreover, the figure legend in Fig S6 states that the plot shows "Log 10-fold change LFQ", but the x axis of the plot is labeled "LFQ intensity". Additionally, the pan Ras staining in Fig S3 shows two bands in gel 1 and three bands in gel 2, whereas in Fig S2 and Fig S3 there is one primary band when exogenous KRAS is expressed, which is confusing. Although the legend of Fig S3 indicates that the additional bands are HRAS and NRAS, why are these additional bands not shown in the other gels? I realize that transient transfections are inherently variable, and you mention this in the discussion, but I think acknowledging that the exogenous FLAG-KRAS expression is dependent upon culture conditions, correlated with the number and content of enriched KRAS effectors, and that the phenotypic effects could be associated with abundance rather than complex composition, is warranted. Thus, modifying claims throughout the paper to acknowledge this relationship would be necessary. Alternatively, if exogenous KRAS expression under the various culture conditions prior to enrichment via anti-FLAG staining of the western blot (Fig S3A) or unenriched mass spectrometry (Fig S4) does not show any correlation with the effector enrichment or content, disregard these comments. Either way the data are informative, but this experiment significantly impacts interpretation of the results and the claims of your paper."

Reply to comment 4: The pan-Ras antibody does detect the FLAG-KRAS protein as it runs higher on the gel used for western blotting than the endogenous Ras protein. Indeed, using a pan-Ras antibody enables to visualize both, exogenous and endogenous Ras on the same western blot and enables to directly quantify the amount of exogenous vs endogenous Ras. While this was already indicated on Fig S3, we also mention this now in the figure legend of Fig S3 that the highest band corresponds to the exogenous FLAG-KRAS band: "*The exogenous Ras band can be distinguished from the endogenous Ras bands as the FLAG tag makes KRAS slightly bigger and therefore this band has an upshift.*" We also clarify this now in the legend of Fig. S2: "*Using the pan-RAS antibody, both the exogenous KRAS band and the exogenous Ras band is seen as the FLAG tag makes KRAS slightly bigger and therefore this band has an upshift.*"

Regarding the detection of either 1 or 2 endogenous Ras bands, we have observed those bands sometimes, but not always. It seems to depend on the actual percentage and run time of the gel prior to the western blot analysis. In fact, a second band begins to appear on gel 1 of Fig. S3, too. I have previously tried to understand where the additional band comes from; some people suggest it comes from the difference in GTP/GDP load and thus charge changes; other people suggest it might be related to posttranslational modifications. We have added to the caption of Fig. S3 that sometime endogenous Ras appear as a double band: "*The endogenous Ras band, depending on the gel run time, sometimes shows up as double band.*"

With respect to the variability in FLAG-KRAS expression, as suggested by the reviewer, we have performed a correlation analysis of the KRAS bait abundance (Fig S6) with the number of proteins enriched in each cell culture condition (Fig. 1A) (see Figure 1 for reviewer below). There is no correlation. Hence, in this case – as the reviewer suggests – there was no need to modify any claims in the paper.

Nr of proteins in complex (Fig 1A)

With respect to the consistent representation of protein abundances, we have modified Fig. S6 so that it has a similar style as Fig. S4 and in both cases the log₂ LFQ intensities are shown. We have altered the caption of Fig. S6, too. As a result, all mass spectrometry data are now shown consistently as log₂ LFQ intensities.

For the western blot quantifications, we prefer to use non-log data, as this is generally how western blot data are represented.

With respect to Ras isoforms, although the pan Ras antibody can detect HRAS and NRAS, too, in colorectal cancer cell line KRAS would normally be the dominant variant, not NRAS or HRAS (see <https://pubmed.ncbi.nlm.nih.gov/36470426/>).

Original comment/response

Comment 4: "Claim 4: "All together, this supports our initial hypothesis that group 2 effectors tend to be found in complex with Ras only in specific conditions that promote PM recruitment via RBD- independent mechanisms."

This is supported by computational analysis. Experimental validation showing increased PM colocalization via microscopy in Caco-2 cells would better support this claim."

Response 4: In the experimental AP-MS setup used in this work, exogenous expressions of Flag-KRAS WT and mutant variants are used. Techniques that assay protein-protein interactions based on co-localization use endogenous proteins that are detected using antibodies, for example proximity ligation assays (PLA). While AP-MS experiments can be performed in a (semi)-high-throughput fashion, PLA assays cannot as high quality antibodies are needed and substantial experimental optimization is needed on a case-by-case (for each effector) basis. Furthermore, PLA is typically not performed on exogenously expressed proteins as it is difficult to account for cell to cell expression variation. Nevertheless, we have performed PLA assays on WT Caco-2 cells and the effector PI3K with the stimulus EGF,

where we show increased Ras/PM co-localization, which however changes over time. We use these results in the discussion to highlight pros and cons of different methods for assaying protein-protein interactions.

Comment 5: *“Reviewer 1 response to comment 4: The PLA analysis seems to support your hypothesis, but relevant controls are lacking. For example, a negative control treatment in addition to the EGF stimulation over the time course. You imply that this control was included in the figure legend for S18, but do not show this data. Without this control, you cannot tell whether the signal is background variation over the timecourse.”*

Reply to comment 5: We understand the reviewers concern, however, we argue that it is highly unlikely that the signal should unspecifically be changed due to EGF treatment. We have empirically observed, developing proximity methods, that background remains unaffected by ligand treatment. The background signal that might appear comes almost exclusively from primary antibodies, and is unchanged in presence or absence of ligands. The contributing authors from Uppsala have a long experience from developing RCA based proximity methods, such as UnFold PLA that later became commercialized under the brand name NaveniFlex (PMID: 29599435) and MolBoolean (PMID: 35963857).

We were not ourselves able to find any formulation specifically implying that negative controls were included, but will re-formulate, if such should be pointed out to us, to avoid any unnecessary confusion.

Comment 6: *“General Comments:
There a many grammatical errors that need to be fixed in addition to the claims about culture context dependent KRAS complex composition. Specifically, in the results section and the discussion. Although this may be a language difference, in general, "mass Spectroscopy" is no longer used to describe "mass spectrometry" experiments, and should be changed throughout the paper.”*

Reply to comment 6: We have changed “mass spectroscopy” to “mass spectrometry” and corrected grammatical errors (changes highlighted in blue in the main manuscript text).

February 22, 2023

RE: Life Science Alliance Manuscript #LSA-2022-01670-TRR

Prof. Christina Kiel
University of Pavia
Molecular Medicine
Via Forlanini 6
Pavia, Lombardia 27100
Italy

Dear Dr. Kiel,

Thank you for submitting your revised manuscript entitled "Analysis of context-specific KRAS-effectors (sub)complexes in Caco-2 cells". We would be happy to publish your paper in Life Science Alliance pending final revisions necessary to meet our formatting guidelines.

- please upload your supplementary figures as single files and add the supplementary figure legends to your main figure legends section
- please upload your table files as editable doc or excel files
- please add the Twitter handle of your host institute/organization as well as your own or/and one of the authors in our system
- please make sure that the author order in the manuscript and our system match
- please add figure callouts for Figure 5B, Figure S11 B & C and Figure S14 B-F and S14H to the main manuscript text
- datasets PXD035399 and PXD039404 should be made publicly available at this stage

Figure Check:

- please add sizes next to the blots in Figure S3A

A. FINAL FILES:

B. MANUSCRIPT ORGANIZATION AND FORMATTING:

Sincerely,

February 27, 2023

RE: Life Science Alliance Manuscript #LSA-2022-01670-TRRR

Prof. Christina Kiel
University of Pavia
Molecular Medicine
Via Forlanini 6
Pavia, Lombardia 27100
Italy

Dear Dr. Kiel,

Thank you for submitting your Research Article entitled "Analysis of context-specific KRAS-effectors (sub)complexes in Caco-2 cells". It is a pleasure to let you know that your manuscript is now accepted for publication in Life Science Alliance. Congratulations on this interesting work.

DISTRIBUTION OF MATERIALS:

Again, congratulations on a very nice paper. I hope you found the review process to be constructive and are pleased with how the manuscript was handled editorially. We look forward to future exciting submissions from your lab.

Sincerely,
